

# Technical Note on high-frequency, multi-elemental stream water monitoring: experiences, feedbacks, and suggestions from seven years of running three French field laboratories (Riverlabs)

Nicolai Brekenfeld[1], Solenn Cotel[2], Mikael Faucheux[1], Colin Fourtet[2], Yannick Hamon[1], Patrice Petitjean[3], Arnaud Blanchouin[4], Celine Bouillis[1], Marie-Claire Pierret[2], Hocine Henine[4], Anne-Catherine Pierson-Wickmann[3], Sophie Guillon[5], Paul Floury[6] and Ophelie Fovet[1]

[1] INRAE, Institut Agro, UMR SAS, Rennes, 35042, France
[2] ITES Institut Terre et Environnement de Strasbourg, CNRS/Université de Strasbourg, Strasbourg, 67000, France
[3] Géosciences Rennes UMR CNRS6118, University Rennes, Rennes, 35042, France
[4] University of Paris-Saclay, INRAE Jouy-en-Josas - Antony, UR HYCAR, Antony, 92761, France
[5] Centre de Géosciences, MINES Paris, PSL University. Fontainebleau, 77300, France
[6] Extralab SAS, France

*Correspondence to*: Nicolai Brekenfeld (nicolai.brekenfeld@gmail.com) and Ophelie Fovet (ophelie.fovet@inrae.fr)

**Abstract.** High-frequency and multi-elemental stream water monitoring are acknowledged as necessary to address data limitation in the fields of catchment sciences and freshwater biogeochemistry. In recent years, the development of stream bank analyzers and on-site field laboratories to measure various solutes and/or isotopes at sub-hourly measurement intervals is in progress at an increasing number of sites. This trend should likely persist in the future. Here, we share our experiences of running three French field laboratories (called Riverlabs) over seven years. This technical note gives an overview of the technical and organizational points that we identify as critical in order to provide guidelines for the successful implementation of future projects running such equipment. We therefore share the main stages in the deployment of this tool in the field, the difficulties we encountered and the procedures we used to identify and eliminate their causes. Some of the critical aspects discussed here relate to 1) Supply of the field laboratory: basic functioning of the pumping, filtration and analytical systems, 2) Data quality control and assurance via maintenance services and operations, 3) Data harmonization and coordination of the laboratory components, and 4) Team structure, skills and organization. Our two main conclusions for a successful, long-term functioning of these types of field laboratories are, first, the necessity to adapt several central components of the field laboratory to the local conditions (climate, section, topography, water turbidity, power) and, second, the need of diverse and in-depth technical skills within the engineering team. We believe that sharing these experiences, combined with providing some practical suggestions might be useful for colleagues, who are starting to deploy such or similar field laboratories. These considerations will save time, improve performance and ensure continuous field monitoring.



## 1 Introduction

Over the last two decades, monitoring of water quality parameters at high temporal frequencies has strongly developed based
on various technologies (Wade et al., 2012; Rode et al., 2016b; van Geer et al., 2016; Bieroza et al., 2023) with the conviction
that concentration data sets with increased resolution will be the key of advancing environmental sciences (Kirchner et al.,
2004; Kirchner et al., 2023). Among these technologies, the emerging field laboratories, so far mainly used for surface waters
as riverbank side analysers, can be qualified as the most sophisticated technologies (Jordan and Cassidy, 2022). They consist
of running chemical analytical instruments in the field in order to avoid some disadvantages of the sample storage and the
delay between sampling and analysis. For some solutes, field laboratories are the only possibility to measure in-situ their
concentration because no other, simpler analytical solution exist. Such technologies are associated to technical challenges that
are related to the acceptable sample filtration, to the required level of maintenance and electrical power source (Bieroza et al.,
2023). Jordan and Cassidy (2022) distinguished wet chemistry analysers, that are generally adapted from those used in the
water utility or water treatment industries, from laboratories developed specifically for the use in the research sector. The latter
include equipment of analytical laboratories such as laser spectrometers for isotope analysis (von Freyberg et al., 2017) or
ionic chromatography for anion and cation analysis (Floury et al., 2017). Bieroza et al. (2023) identified six topics where
significant advancements were achieved thanks to high-frequency water quality monitoring: (1) the scaling-up of solutes and
particles with flow (*e.g.* Rozemeijer et al., 2010), (2) the interactions between catchment versus stream processes (*e.g.* Winter
et al., 2021), (3) the quantification of stream metabolism (*e.g.* Rode et al., 2016a), (4) the quantification of global change
impacts (*e.g.* Speir et al., 2021), (5) the provision of evidence for management decision (*e.g.* Skeffington et al., 2015), and (6)
the improvement of measurement and modelling (*e.g.* Tunqui Neira et al., 2020).

However, to our knowledge, there is little literature from a technical and operational point of view, which can be used by the
scientific communities that would like to design or run such analysers. Nevertheless, improving the quality of high-frequency
data sets was emphasized by Bieroza et al. (2023) as a critical issue, associated to the urging need for the development of
robust protocols for maintenance and data management. Therefore, our objective of the present technical note is to provide
guidance for running field laboratories, by detailing critical technical points to achieve as soon as possible continuous, reliable
and usable datasets. We do this by sharing our experiences from the three Riverlabs and by proposing some suggestions that
we successfully tested. These "Riverlabs" are prototypes of geochemical laboratories in the field developed within the CRITEX
project: Challenging equipment for the temporal and spatial exploration of the Critical Zone at the catchment scale (Gaillardet
et al., 2018), an instrumental program aiming at sharing innovative analytical facilities for French research in geosciences.

The note is structured into four parts: the water supply and filtration (section 3), the interactions between the components
(section 4), including general components (4.1) and measurement technologies (4.2), the maintenance (section 5), including
internal maintenance (5.1) and that provided by suppliers or manufacturers (5.2) with associated frequencies, and, finally, the



organization of running the Riverlabs (section 6). We usually illustrate encountered issues with illustrations and then present our developed solutions.

## 2 Material and Methods

### 2.1 Study sites

Three French sites were equipped with a Riverlab (Endress+Hauser, France), which are all part of Long-term Critical Zone Observatories (OZCAR-RI), distributed from Western to Eastern France (Figure 1). The Avenelles catchment was equipped in 2015 with a first version of the prototype, whereas the Strengbach and Naizin catchments were equipped in 2017 with a second version of the prototype. We describe briefly the three sites from west to east.

The **Naizin** catchment (AgrHyS Observatory, Fovet et al., 2018) covers 5 km$^2$ in central Brittany. The elevation ranges from
98 to 140 m a.s.l. and slopes are gentle (<5% on average). The bedrock is composed of upper Proterozoic schists. The rather impervious unaltered bedrock is overlain by a fractured zone, where water can percolate, and by a weathered layer, 1 to 30 m deep, where shallow groundwater can fluctuate. The climate is temperate and humid with an average rainfall of 837 mm and an average temperature of 11.2°C. The soils are silty loams, between 0.5 and 1.5 m deep, and well drained except in bottomlands close to the streams. Land use is dominated by agriculture (91%), with cereal crops (maize, straw cereals) and
grasslands.

The Avenelles catchment is a 45 km$^2$ sub-catchment of the **Orgeval** experimental catchment (ORACLE Observatory, Tallec et al., 2013) located 70 km east of Paris. Its elevation ranges from 85 to 130 m a.s.l. and the topography is relatively flat. The geological context corresponds to a multi-layer aquifer system with two sedimentary tertiary formations: the shallower Brie aquifer (Oligocene limestone) that is separated by a discontinuous grey clay layer (Priabonian mudstone and Bartonian marl)
from the deeper Champigny aquifer (Eocene limestone). The climate is semi-oceanic, average annual rainfall is 740 mm and mean annual air temperature is 9.7°C. Deep loamy soils are highly homogenous, and usually tile-drained (80 % of the agricultural lands). Land use is dominated by agriculture (82%), with cereal crops (wheat, maize, barley and pea).

The **Strengbach** Catchment (OHGE observatory, Pierret et al., 2018, 2019) is a 0.8 km² granitic watershed located in the Vosges Mountains. Its elevation ranges from 880 to 1150 m a.s.l. with heavily incised side slopes (mean slope of 15°). The
bedrock is mainly composed of Hercynian Ca-poor granite (315 ± 7 Myr), which was subjected to hydrothermal alterations of various levels with main hydrothermal alterations occurring 183,9 My ago (El Gh'mari, 1995). The thickness of the granite arena varies from 1 to 9 m. The local climate is temperate oceanic mountainous with a mean annual temperature of 6°C. Average annual precipitation is 1380 mm, with snowfall occurring 2 – 4 months per year. The soils are brown acidic to ochreous podzolic series and are generally about one meter thick. Land use is dominated by forests (85%), composed mainly
of spruces (80%) and beeches (20%).



## 2.2 Riverlab prototypes

The three Riverlabs have an almost identical design and functioning and show only minor differences (for further details about the first Riverlab see (Floury et al., 2017)). The first Riverlab, installed in the Orgeval catchment in 2015, is slightly different

from the two other Riverlabs installed in the Strengbach and Naizin catchments in 2017, leading to a higher similarity between the two latter ones. In general, the Riverlabs can be divided into three main parts, namely the pumping system, the filtration system, and the analytical instruments and sensors, which are housed in a container-like shelter.

A surface pump, located inside the Riverlab, provides a continuous flow of unfiltered stream water, with a regulated, constant flow and pressure. The stream water is aspired through a strainer in the stream (variable sizes and configurations for the

different Riverlabs) and pumped through a hose (inner diameter ~2 cm; length of around 5 m at Naizin and Orgeval and 141 m at Strengbach) into the Riverlab at a flow rate of 500 to 1300 l/h. Inside the Riverlab and after the pump, the unfiltered water flows through flow and pressure sensors, a tangential filter, an adjustable pressure valve, and an overflow tank with sensors for physico-chemical parameters. Sensor probes used are for temperature Thermophant T TTR31 (Endress+Hauser), for electrical conductivity (EC) Condumax CLS21D (Endress+Hauser), for dissolved oxygen (DO) Oxymax COS61D

(Endress+Hauser), and for pH Orbisint CPS11D (Endress+Hauser). At the Strengbach catchment, the unfiltered water is also flowing through a turbidity sensor (Turbimax CUS52D, Endress+Hauser). The regulation of the flow and the pressure of the unfiltered water is achieved manually at the Orgeval Riverlab whereas at the other two sites this is done automatically with a PID (proportional integral derivative) controller. The controller is continuously adjusting the aperture of the adjustable valve and the pump speed (via a variable-frequency drive) depending on the differences between the measured and targeted water

pressure and flow, respectively.

The filtration system is composed of two parts. In a first part, the unfiltered, pressured stream water is flowing through a tangential, stainless steel filter (pore size of 0.5 to 2 µm, depending on the site), which is continuously producing filtered stream water (with a flow rate between 10 and 15 l/h at Strengbach and 1.5 l/h at Naizin, not measured at Orgeval). This first tangential filter is cleaned automatically every few minutes (ultrasound and reverse flow) and, additionally, manually every

few weeks. Around two thirds of this filtered water is flowing through DO, EC, and pH sensors (at Strengbach) and is used for further analysis in other instruments (LAR QuickTOCuv from Anael for dissolved organic carbon (DOC) and Stamolys CA71SI from Endress+Hauser for dissolved Si at Strengbach and Naizin). A small part (a few percent or less) is filtered through a second paper filter (0.22 µm) in a spiral tangential filter set-up (Strengbach and Naizin) or in a frontal cylindrical filter set-up (Orgeval) to provide filtered water to the ion chromatography system (ICS5000+ from ThermoFisher Scientific).

This second filter is replaced manually every week to fortnight.

All analytical instruments (ICS, DOC analyser, Si analyser) and sensors (DO, EC, pH, turbidity) are regularly checked for drift, cleaned, and calibrated. The chemicals and pure water, that are required for the analyses by the instruments, are regularly prepared in the laboratories and subsequently delivered to the Riverlab. The effluents and liquid wastes produced by the





instruments are either ejected back into the stream, if they are of no environmental concern, or collected in containers and

subsequently returned to the laboratories for appropriate processing.

In the following sections (3 to 6), we present some of the challenges we encountered, the solutions we applied and some practical guidance for future users of this or similarly complex field laboratories.

## 3 Water supply of the field laboratory

### 3.1 Stable supply of unfiltered stream water during storm events and variable hydrological conditions

During (high) rainfall events, water level, turbidity and particle concentrations increase in the watercourses, sometimes by several orders of magnitude (Cotel et al., 2020; Vongvixay et al. 2018; Lefrancois et al., 2007). The variations can be very rapid, especially in very small headwater systems or during events such as flash floods. In general, these events can lead to an increased presence of bubbles and vortexes in the stream. Moreover, in temperate catchments, the autumn season is associated with a flushing of small fishes, tree leaves, branches and other clogging material that can block the opening of the strainer. All

these disturbances can reduce the stability of the pumped flow delivered to the field laboratory and likely affect the residence time of the water in the water circuits.

### 3.1.1 The supply of unfiltered water

Based on our experience, we identified four major potential problems of malfunction and failure of the pumping system. These are a) the choice between a surface or submersible pump, b) the design of the strainers, baffles and the intake hose, c) the

combination of the pump and the variable frequency drive, and d) the appropriate regulation of the pump by the PID controller. Regarding the pump type, all three Riverlabs were equipped with a surface pump (Wilo, WJ-203-X-DM-IE3 or Grundfos, JP-5-B-A-CVBP-E-N at Naizin or Strengbach). However, we tested a submersible pump for around five months at Naizin (Calpeda, borehole pump, type: 4CS 0,55T – 4SDX 1-9). Pump suppliers often advocated for submersible pumps because they do not need to be primed and because pushing the water is often easier than aspiring it (with a surface pump). This is especially

the case if the surface pump is located more than a few meters above the water intake. However, our surface pumps almost never lost their prime even if they were stopped and re-started remotely. In contrast, one of the observed disadvantages of the submersible pump was related to its size. At Strengbach (80 ha), the stream is too small to allow the installation of a submersible pump and even at Naizin (5 km$^2$), the submersible pump occupied an important portion of the stream cross-section. Another disadvantage is the exposure to mineral and organic material, deposited into the pump body. We tried to prevent inhalation of

debris with baffles and strainers. However, deposition of fine organic and mineral particles into the submersible pump led to strong wear and tear already after a few months (Figure SI 1).

During storm events and independent of the pump type, the pump was frequently not able to supply a stable water flow close to the target value to the Riverlab at Naizin, likely due to the high concentration of fine particles (Figure 2). Another reason for the instability of the water supply during storm events was related to the strainers and baffles, which were, initially, not



adapted to the local conditions, and frequently clogged during storm events. At Naizin, with important inputs of large
       broadleaves and branches in autumn, the initial strainer was too small and therefore frequently clogged by a few leaves during
       the first autumn storms. The strainer was then adapted manually and eventually consisted of a small strainer with a small pore
       size nested within a larger strainer with larger pores, which has been working well since. At Strengbach, in contrast, the initial
       rectangular strainer was too large for the cross-section of the stream and therefore accumulated debris during storm events,

but, at the same time, had too few and too small pores. A cylindrical ISCO-type strainer (ISCO automatic sampler, Neotec
       Ponsel) was then used, which was additionally equipped with an easily removable mesh around it. After several trials of
       different mesh-sizes, a size of 2 mm was chosen and has been working successfully since (Figure 3).

       A variable frequency drive is controlling the pump speed and needs to be chosen in function of the pump properties and
       following the recommendations of the suppliers (we used Eaton Industries, DC1-127D0FN-A20N). In case a pump needs to

be changed, the compatibility of the variable frequency drive with the new pump needs to be evaluated or changed, if necessary.
       Equally, the variable frequency drive needs to be checked for damages and malfunctioning regularly, as it can be the hidden
       cause of erratic pump failures. For example, at Naizin, we observed very irregular and more or less frequent pump failures
       during baseflow periods (Figure SI 2). Only after a long diagnostic phase, we realized that a malfunctioned variable frequency
       drive was the primary cause.

Finally, the PID controllers can be adjusted by tuning some parameters to optimize the response to a deviation of the measured
       flow and pressure of the unfiltered water from its target values. It is important to establish different adjustment times of those
       two controllers (for the flow and pressure, respectively) in order to avoid instabilities. For example, we set a relatively fast
       adjustment of the flow and a slow adjustment of the pressure. In addition, both adjustment times need to be slower than
       instabilities of the whole unfiltered water system that occur especially high during storm events. It is inadequate to have a fast-

reacting PID controller that is trying to adapt to the instabilities of the system. The parameters of the PID controller need to be
       established carefully on a case-by-case basis. For example, at Naizin we used values for the proportional and integral
       parameters (no derivative parameter was used) of 0.3 [-] and 5.0 [s] for the flow and of 0.2 [-] and 50.0 [s] for the pressure,
       respectively. However, since we do not know the exact formula, which uses these parameters in combination with the flow
       and pressure measurements, the absolute parameter values might not be very informative for other users, which use different

PID systems.

### 3.1.2 Tips and good practices

Following the main challenges concerning the pumping system in the previous section, we provide here some guidance for a
successful design of this system and some potential points of attention. From our experience, an individual adaptation of the

pumping and strainer system to the local conditions is paramount. For this local adaptation, important characteristics to
       consider are: turbulence and sediment transport capacity of the stream, particle size distribution of the suspended sediments,
       input of organic debris, stream width and depth, as well as maximum and minimum water temperature (risk of freezing).



Commercially available and adapted or hand-made strainers in addition with baffles or deflectors are necessary in order to prevent the clogging and the wear of the pump. However, there is no one-fits-all design of the strainers. The pore size of the

strainer needs to be large enough to allow the provision of a maximum flow and at the same time, it needs to be small enough for protecting the pump and filtration system from too much debris and sediment. We advise to locally conduct some tests with different pore sizes in different hydrological and turbidity conditions. Finally, the strainers should not be too close to the streambed in order to avoid the influx of bedload particles.

The type and number of intended analyses and their analytical requirements in terms of water volume (for the analysis and the

turn-over of the water volumes) might also impact the required flow and, therefore, the choice of the pump. Additionally, the altitude difference between the field laboratory and the water intake needs to be considered when choosing the pump type, as explained above. At Orgeval and Naizin, the water intake is a few meters below the field laboratory while at Strengbach, a long hose between the strainer and the Riverlab is responsible for an inverse gradient, where the water intake is above the field laboratory. In that latter case, the challenge is less to prevent pump failures but more to conduct maintenance of the long hose.

Furthermore, locations with freezing winter temperatures might require specific water evacuation systems to avoid freezing of water in some slow flowing parts of the system. At Strengbach, for example, a temperature-regulated valve opens and drains the whole pipe system of the field laboratory, if the water temperature drops below a certain threshold. In addition, a small part of the intake hose is heated, whereas the remaining part has been buried into the streambed in order to reduce the risk of freezing. Finally, it is very advisable to have remote access to the field laboratory and its main components with the possibility

of re-starting the pumping system remotely.

### 3.2 Stable supply of filtered stream water

The filtration of water before analysis usually meets two different objectives. One is related to the scientific questions and interpretations of the analyses when one targets the quantification of dissolved phases of some constituents. This is the case,

for instance, for the dissolved forms of some nutrients (P, N, Si). Another motivation for filtering is to prevent the clogging of the circuit and the wear of the analytical equipment. However, any filtration introduces bias in the analysis (Horowitz et al 1992). Finally, when designing the on-line filtration system, a trade-off needs to be found between the risk of filter clogging (due to a high flow through the filter) and the risk of a long residence time in the circuits (due to a low flow through the filter).

### 3.2.1 Critical aspects and challenges of the filtration system

Filtration of the water must take into account the scientific aspects, as the choice of measuring dissolved versus total or particulate species is scientifically driven, but also technical criteria. Indeed, filtration is a compromise between the addition and management of a filtration unit and the preservation of analytical units. As an example, in the initial design of the Riverlabs, the carbon analysis (Total Organic Carbon concentration) was carried out on the unfiltered water. However, during high-flow





periods, we observed heavy fouling of the tubing and the analytical equipment. Therefore, we decided to analyse the DOC
concentration on water filtered at 0.5 µm by the tangential filter to preserve the analyser and to avoid clogging of the tubing.

In the Riverlabs of Strengbach and Naizin, the water filtered by the stainless steel tangential filter (0.5 µm) is directed towards
the spiral cellulose acetate filter (diameter of 47 mm and pore size of 0.22 µm) which feeds the ICS. As mentioned above in
section 2.2, the flow of the unfiltered and filtered (0.5µm) water were 600 l/h and 1.5 l/h at Naizin and 500 l/h and 10 – 15 l/h
at Strengbach, respectively. At Naizin, the flow after the second filtration step (0.22µm) varied between 0.03 and 0.18 l/h
(Figure 4) (not measured at Strengbach). Accordingly, the pressure drop across the first filtration step by the tangential filter
was smaller at Naizin (around 0.5 bar, from 1.8 to 1.3 bar) than at Strengbach (around 1 bar, from 2.0 to 1.0 bar). Several
filtration biases were identified and attributed to the 0.22 µm paper filter.

First, the spiral filter with a surface area of only 17.3 cm$^2$ (the area used for the actual filtration is much smaller) tends to clog
at a rate that depends on the hydrological condition and on the size of the suspended solids in the stream. During floods, when
the water is turbid, the flow passing through the 0.22 µm filter decreases very quickly, up to 83% within a week (Figure 4).
This has a significant impact on the residence time of the water within the circuits that is not constant over time. It therefore
leads to asynchronies between the different analytical equipment depending on whether they are supplied with 0.22 µm filtered
water or not (Figure 5). We will come back to this problem of the asynchrony between different instruments in section 4.3. In
addition to the organic and mineral material coming directly from the stream, particles and biofilm can accumulate in the
tubing of the circuits and can be remobilized later during cleaning sequences or surges of water flows (e.g. when replacing the
0.5 µm filter). Hence regular replacement or cleaning of as much plumbing as possible is required, especially through parts
that form angles and that cannot be easily disassembled, such as the oxygen probe and the filtered water tank.

In addition to the effect on the flow-through and residence time (which we will treat in section 4.3), this clogging had a direct
effect on the analytical results of certain elements, particularly sulfate. ICS-analysed sulfate concentrations tend to decrease
during filter clogging (up to 30% at Naizin during one week (Figure 6)). Lab analyses of samples taken from the circuit before
and after the spiral 0.22 µm filter indicated that the filter acted as a sink of sulfate. The removal of sulfate increased over time
and with the clogging state of the filter. This issue of the sulfate measurement was not observed with the frontal cellulose
acetate 0.22 µm filter at Orgeval, where sulfate concentrations are also one order of magnitude higher than in the two other
streams.

Finally, we also identified a contamination problem affecting cation analyses on the three Riverlabs. With the ICS in the
Riverlabs, we observed significant concentrations of Ca, Mg and K in Milli-Q water samples that did not contain these
elements, when verified in the lab (ICP-MS verification). Such contamination developed gradually over time (Figure 7) and
can disturb the validation of the calibration curves, with low points affected by this contamination. The ICS had to be flushed
with many Milli-Q water samples in order to achieve a sufficiently low level of background concentrations.

In Strengbach, the gradual increase of the calcium (+ 250% in a month), magnesium (+ 180% in a month) and nitrate (+125%
in a month) concentrations was confirmed when analysing certified material (a lake water with a chemical signature similar to
those of the Strengbach River) every two weeks, showing that these elements adsorb to or are released by the tubing material.



A similar effect was observed in the other two Riverlabs, but to a lesser extent, likely due to the higher background ion concentrations.


### 3.2.2 Solutions and interpretations

The various tests carried out showed that the sulfates were adsorbed on the paper filter or on the particles retained on the filter, since the sulfate concentration analysed at various points in the filtered water line only decreases at the outlet of the 0.22 µm

filter. Changing the filter material from cellulose acetate to nylon (conducted at Naizin during several weeks) did not solve the problem in terms of both the decrease in flow (clogging) and the loss of sulfate. The decrease in sulfate concentration is all the more visible as the concentration in this element is low. A decrease of 3 mg/l by the filter represents around 30% in Naizin while in Orgeval, this decrease was likely not noticed, because the background sulfate concentration is much higher (70 mg/l), and the particles load to the filter in less important. In addition, the nature of the solid suspended matter retained on the filter

is different depending on the geology of the sites and on the land use of the catchments. Finally, we closed the by-pass circuit of this spiral acetate 0.22 µm filter (that is, we forced all the water through the filter), and the sulfate issue was almost completely solved.

Regular cleaning of all the PEEK tubing, including the injection loops, as recommended by Thermo with acetonitrile/water (1/9) and HCl (1N) solutions, respectively, followed by rinsing with Milli-Q water helps to reduce the contamination and to

find an acceptable background concentration of the blank. The other alternative is to change the PEEK tubing regularly. Although PEEK offers very high chemical resistance to many elements, base acids, or solvents, our observations show that over time this material can, under certain conditions, gradually adsorb certain elements and release them during the passage of corrosive solutions such as ultrapure water or standard solutions made with Milli-Q water. It is crucial that any change or cleaning on the tubing preceding the injection loop requires a new calibration of the instrument and the analysis of certified

material, to be sure that the subsequent injections will be analysed accurately. In Strengbach, the injection loop and the tubing segments are changed at least every month or when a suppressor/column change is made, and a new calibration of anions and cations is performed after stabilization of the signals. Contrary to Strengbach, in the Naizin and Orgeval Riverlabs, we only noticed a slight (Naizin) or no (Orgeval) long-term contamination of the measurements of stream water. This might be due to the higher cation concentrations in the latter two catchments.

## 4 Data harmonization and coordination of the laboratory's components

The field laboratory is composed of many different analysers and sensors with different measurement and acquisition time steps. Therefore, the time series measured by the different instruments have to be synchronized before data analysis, which is not always straightforward. The exploration of synchrony between concentrations or any other parameter measured in the



Riverlabs and the stream flow (C-Q relationships) needs thus to be performed cautiously. In addition, some sensors interfered
with other equipment in the Riverlab. It is, therefore, critical to be aware of potential interferences between different
components of the Riverlabs and to assess the synchrony or asynchrony of the acquisition by the different sensors and analytical
instruments.

## 4.1 Connectivity, interactions, and interferences between the different devices

In the Riverlab at Naizin, the water level measurements were artificially modified by the ventilation system. A vented water-
level sensor was used to measure the stream water level, with the opening of the vent located inside the Riverlab. The sensor
therefore measured the pressure difference between the stream (water level pressure) and the inside of the Riverlab (barometric
pressure). Under normal conditions, the atmospheric pressure inside and outside of the Riverlab is sufficiently similar.
However, a ventilation system, that expelled air from inside the Riverlab to create an airflow, reduced slightly the air pressure
in the Riverlab, relative to the outside pressure. Even though this artificial pressure difference was likely only about 0.1 mbar,
it created an artificial water level increase, based on the vented level sensor, of around 1 mm, which was noticeable especially
during the summer low-flow period. To avoid this problem, we advise to install the opening of the vent outside of a ventilated
field laboratory.

Another example of unexpected interferences was observed between a powerful uninterruptible power supply (UPS) and
differential circuit breakers. Almost the entire Riverlab is connected to an UPS system. Furthermore, the whole electrical
circuit is protected by differential circuit breakers to protect the people working in the Riverlab. However, specific differential
circuit breakers are required that are on the one hand sensitive enough to protect the people and on the other hand robust
enough (i.e., accepting very short surcharges) to cope with an UPS. The initial installation did not consider these aspects,
resulting in sporadic and irregular power outages. Identifying the causes of these power outages was a major challenge for the
local team as well as for the external equipment providers.

Lastly, we encountered frequent, but irregular, disconnection errors of the ICS. Various modules of the ICS (e.g., dual pump,
eluent generator, chromatography module etc.) are connected to a central computer with USB cables. Initially, we used
standard USB cables and a USB-hub. Following the repeated disconnection errors of some of the modules, we replaced the
hub by an internal USB expansion card and the standard USB-cables by shielded cables. These replacements solved our
disconnections errors.


## 4.2 Choice of analytical and sensor technologies

The choice of the analytical and sensor technologies in the Riverlab has consequences for the exploitation of the acquired data.
The choice criteria of the sensors and the analytical instruments depend on: (i) the precision and stability of the technology,
(ii) its suitability to the local chemical conditions, such as the ranges of values in the stream water, and (iii) the consistency
between instruments, mainly with instruments collecting and analysing the sample water differently.



**(i)      Precision and stability of the technology:**

This is related to the compromise between the direct uses of laboratory instruments in contrast to analytical instruments that were specifically designed for field operations. The latter might have a lower quantification performance but also a lower reagent consumption and waste production, as well as a better robustness. In addition, the long-term, continuous operation can

have impacts on the performance of classical laboratory instruments that were not designed for this continuous operation. According to our investigations and to an engineer working at the manufacturer company, the long-term drifts (a few months) of the calcium and magnesium concentrations observed at Naizin and Strengbach was likely linked to the PEEK tubes of the ICS, that adsorb and accumulate certain ions or impurities. For regular use, lab-based, non-continuous measurements, the PEEK tubing might be adequate, but an alternative material in continuously operating field laboratories might be useful.

**(ii)      Suitability to the local chemical conditions:**

In addition to the choices of the sensor technologies and analytical instruments, the analytical protocols, including the calibration and the controls with certified standards, need to be adapted to the local conditions and requirements as well. For instance, at Orgeval, calcium precipitated in the tubes of the Riverlab and instruments (ICS) as well as in the overflow tank and on the sensors. Frequent cleaning and purging with acid or the replacement of the tubes were necessary there. In contrast,

at Strengbach, the low ion concentrations were likely responsible for the faster wear of the pH sensor, which required frequent changes. An evaluation of its suitability for these low ion concentrations prior to its installation would have been advantageous.

**(iii)      Consistency between instruments:**

Different instruments might extract or accumulate their water samples differently, with variable integration time. This has an effect on the synchrony of the measurements. In catchments with a fast changing water chemistry, it might be important to

consider these aspects to facilitate the process of synchronization of the acquired data. The DOC analyser in the Riverlabs at Naizin and Strengbach, for example, uses continuous flow. Its quasi-continuous measurements are an integration over several minutes to tens-of-minutes. This is in contrast to the ICS, where discrete measurements are based on the injection of a small sample volume at a precise and short moment of time. In addition, a variable and unequal transfer time from the stream to these different instruments adds another source of inconsistency. Therefore, combining these different types of data needs to

be performed carefully, because the measurements of the DOC analyser and of the ICS, for example, at an instant $t$ do not correspond to the same parcel of water.

## 4.3 Solutions and perspectives for synchronization

Automatisation of the data qualification can be partially achieved using variables related to the system states (pressure of filtered water, pump speed). For instance, the solution proposed by the Extralab company (Paris, France) was to create filters

for excluding data corresponding to a pump speed value that is not equal to the targeted pump speed (plus and minus a tolerance), or for excluding data corresponding to outlier values using expected ranges which have to be specified by the operators.





For synchronizing the different concentration data as well as the hydrological and physico-chemical parameters several non-exclusive options are possible. At Naizin, for example, temperature and/or conductivity measured by sensors in the overflow

tank were also measured directly in the stream, next to the strainer, using a similar sensor technology, which enabled us to quantify and correct the delay between both signals. In Naizin, constant-rate injections of salt solutions at low flow (6.7 L.s$^{-1}$) allowed us to quantify this delay between those two EC sensors as ca. 8 minutes. A similar delay was computed at Strengbach. The same injection allowed us to estimate that the delay between the ICS and the stream can reach up to 30-40 minutes at this low flow value and when the 0.22 µm cellulose acetate filter was one-week old (Figure 8). Estimates of this delay between the

ICS and the stream or the overflow tank, where the EC sensor is located, can also be estimated by computing the theoretical conductivity based on the ion concentrations from the ICS and to compare it to the electrical conductivity measured with the sensor in the overflow tank or elsewhere (Figure 5). This method works especially well, if the EC measurements vary considerably during a storm event. In contrast, this method cannot be used during small storm events, which do not lead to a clear variation of the EC signal (as we observed it at Strengbach). Alternatively, the outflow of the ICS could be measured

continuously and could be correlated with measurements that quantify this time lag. For example, salt injections could be used to measure the time lags between the intake of the water at the strainer and the arrival at the ICS, at various degrees of the filter clogging (new vs. old filter). These measurements could then be used to establish a correlation between the time lag and the measured outflow of the ICS.

Once these delays are estimated, one option might be to average and homogenize the data time series rather than using the

instantaneous ones, for example for the ion and carbon concentrations, pH, EC, DO, temperature and stream water level. Hourly average, for instance, would be still sufficient to study dial variations, and storm flow variations in Naizin and Orgeval. It would be too coarse for storm flow variations in Strengbach though. Instead of a continuous filtration, another alternative in the system design for insuring synchronization would be to filter only a small, sampled volume of water at 0.22 µm at specific and known moments of time as it is done in the Swiss field laboratory (von Freyberg et al., 2017). This choice minimizes the

clogging and the degradation of the filter, but requires an additional sampling step.

## 5 Data quality control and insurance by regular maintenance

As any classical laboratory, the Riverlabs need a quality control procedure and regular maintenance to allow the exploitation of reliable data and metadata. This section reviews the main categories of the maintenance operations that we concluded to be mandatory in our cases, in order to help future potential users to design and anticipate this maintenance part. Even if some

detailed operations will vary depending on the specific settings of the field laboratories (the choice of the equipment, the station's configurations, etc.) we believe that this list might serve as a valuable point of departure to estimate the costs and skills required for running any field laboratory. As an order of magnitude, the operating costs were between 10.000 and 20.000 euros per year per Riverlab in 2018-2022.



## 5.1 Annual to weekly maintenance operations and strategies

### 5.1.1 Major maintenance categories and frequencies

The three Riverlabs are located between 80 and 100 km from their reference research centres. The technical staff performs maintenance operations on site every week (Naizin) or every two weeks (Strengbach, Orgeval). The Riverlabs require very regular maintenance, including cleaning of tubing, pipes, tanks, and sensors, calibration of probes and analytical tools, replacement of filters and consumables (e.g., eluent cartridges, columns, reagents), reparations of various failures, removal of

chemical wastes, quality control using certified standards, as well as purging of the air compressor (Table 1). For these regular maintenance operations, interventions every week or every two weeks seem to be sufficient, in our experience. However, with this frequency of field visits, it can take several months before some sporadic or unknown problems can be properly solved (pump failures, power outages, measurement drift, contaminations, etc.). In addition to the regular maintenance, occasional breakdowns require timely on-site interventions (clogged strainers, pump breakdowns, electrical problems, breakdowns in

measuring equipment, leaks, etc.). For these sporadic instances, local inhabitants were trained to conduct basic interventions, for which they were compensated for. The budget necessary for running a Riverlab has two aspects: the routine and the unexpected reparations. The budget for the routine includes varies expenses and is as complex as an analytical laboratory because of the large number of service providers. The budget for the repairs of large breakdowns is more difficult to plan, especially because the Riverlabs are exposed to a variety of meteorological conditions as well as to human and animal

activities. These repairs included expensive parts of the analytical and technical equipment due to multi-year wear, lightning strikes or others.

Table 1: List of maintenance operations for each component of the Riverlab, with an indication of the frequency of each

operation.

| | Weekly or fortnightly (or after heavy rainfall events) | Intermediate | Annual |
|---|---|---|---|
| Pump | Strainer cleaning | | Replacement or cleaning (prevention of clogging) |
| Small tubing / filtered water circulation | Verification/ cleaning | | Changes twice a year |
| Pipes | Verification / cleaning | Cleaning of removable segments | Replacement or cleaning . |





| Filters | Replacement of 0.22 µm filter and cleaning of 0.5 µm filter. | | Check of performance: if necessary replacement of 0.5 µm filter |
|---|---|---|---|
| Compressor | Purge | | Cleaning and revision |
| Air conditioner | | | Cleaning, filter change and annual control |
| Electric board and circuit | | | Regulatory control |
| Processes (overflow tanks, regulation of water circulation and distribution) | Verification, cleaning | | Replacement, if necessary |
| Sensors (pH, conductivity, O₂, temperature) | Cleaning, verification and calibration if necessary | | Replacement of consumable items (pH probe, O2 membrane), calibration |
| Turbiditymeter | cleaning | | Calibration |
| Analysers | Standards and quality check (contamination, signal shift…) Supply with chemical reagents Daily check of operation | Change of consumable items (eluents, reagents) Calibration, Preventive maintenance | Change of consumable items (columns, tubing) Intervention of manufacturer |
| Pure or ultra-pure water | Refilling of tanks Checking of water quality | | Replacement of cartridges, cleaning |
| Chemical wastes | Collection and removal | Collection and removal | |
| Online Notebook | Registration of all operations and interventions | | Archiving the notebook |

## 5.1.2 Possible alternative strategies for simplifying water supply and chemical waste management

In an early stage of the development phase of the Riverlabs, in the Orgeval observatory, ultra-pure water for the ICS was produced on-site directly from stream water instead of bringing demineralized water to the field for supplying ultra-pure water. However, due to very fast saturation and clogging of the on-site ultra-pure water production system, it was replaced by a



routine, where demineralized water from the laboratories was frequently transported to the Riverlabs. In this updated version of the Riverlab, the demineralized water was stored and subsequently purified by a Millipore System for eluent production and regular purging, which consumed on average around 1 l per day. Since the containers for storing the demineralized water are relatively large (in total 20 l at Naizin, 30 l at Strengbach, only 10 l at Orgeval) they only need to be replenished every 2 to 3

weeks (every 10 – 15 days at Orgeval), which is less frequent than the regular interval at which the Riverlabs are visited. As a side note, the Riverlab at Strengbach was running without a Millipore System for several years, without any worsening of the analytical quality, which reduced the running costs. Ultrapure water (18.2 MΩ) was transported to and stored in the Riverlab and was directly used by the ICS.

In addition to the provision of the reagents for the different analytical instruments, it is very important to consider the handling

of the produced waste. Two aspects are important here: the potential environmental hazard of the waste concentration and its produced volume. For the first case, we evaluated, whether the residual concentrations of the reagents in the waste might reach concentrations that could be of environmental concern, especially during low-flow periods. We concluded that the ejection of the waste coming from the DOC analyser back into the stream was of no concern. In contrast, the waste of the ICS had to be collected. The volume of the produced waste by the ICS (6 l per week at highest analysis frequency) is relatively small and

can therefore be easily handled, by returning it frequently to the lab. This is in contrast to the Si analyser, which is producing up to 35 l of waste per week at the highest sampling frequency (every 20 minutes). We therefore had to reduce the sampling frequency of the Si analyser in order to be able to manage its produced waste. This example illustrates the importance of considering the waste production during the selection procedure of the different analytical instruments, because a low waste production is not always targeted, when fabricants and suppliers develop new instruments. However, a good example, from

our experience, is the Phosphax Sigma Phosphate Analyzer (Hach, UK) (not part of the Riverlab), which has been specifically developed for remote sites with a low waste production of only around 10 l per months with a sample analysis of every 15 minutes.

## 5.2 Support and maintenance by manufacturers and suppliers

The Riverlabs running depends on the correct functioning of many different technical and analytical components, which were

provided by separate suppliers. Due to this unique combination of the different components and the problems arising from their interaction, the individual suppliers were not aware of those problems and their causes, and frequently pointed at components of other suppliers. This made some failures very complex and difficult to solve, and required a competent technical team with detailed knowledge of the local Riverlab. As an example, the above-mentioned bias to the water level measurements, caused by the ventilation system of the Riverlab, was discovered at Naizin. As another example, at Strengbach, we hypothesize

that the air compressor contaminated the pressurized air, which is needed for the DOC analyser and which, therefore, produced erroneous measurements.



For sophisticated analytical instruments, such as the ICS, we valued to have a maintenance contract with the supplier, as it allowed to receive quick and detailed feedbacks, and, therefore, to solve unknown problems more swiftly. This comes, however, with an elevated cost (7.700 euros per year in 2016), which should be factored in from the beginning of the project. However, maintenance costs can be reduced elsewhere. At Strengbach, for example, cartridges of one of the eluents were regularly prepared in the laboratory from a concentrated solution and were not acquired in a ready-to-use concentration from the supplier. This required some additional careful preparation but reduced the cost for the eluent by a factor of almost 50.

**5.3 Consequences for the management of measured time series**

Producing long time series so that they are easily accessible and usable afterwards, often requires a mix of manual and automatic quality control procedures. In our experience, a detailed, technical knowledge of the effect of the different components and their interactions on the measurements is additionally required.

The data validation of the ion concentration time series from the ICS were conducted regularly and manually by technical members of staff, who were maintaining it and who were, therefore, aware of any relevant malfunctions or inaccuracies of the system. This allowed us to directly and undoubtedly remove non-validated measurements from the time series.

In other cases, specific routines in the functioning of the Riverlab had impacts on certain measurements during specific periods. It required detailed technical knowledge in order to interpret these additional variations correctly. For example, a programmed flushing of the whole pumping cycle artificially increased and decreased the temperature in the overflow tank slightly during the flushing. This rapid temperature variation in turn influenced the temperature-corrected electrical conductivity due to a lag of the temperature adaptation of the conductivity sensor.

Interventions in the Riverlab, such as maintenance work, modifications and reparations were all noted down in electronic (and at times paper) format, which was shared and accessible by all project participants. Depending on the noting system, these interventions were, however, not directly linked with the database nor were they in a tabular format. It was therefore not directly possible to visualize the data from the logbooks, such as the days when a specific filter was replaced, in a concentration time series, for example. This type of information had to be extracted manually from the logbooks. In addition, due to the multitude of the work conducted during a single field day at the Riverlab, an individual log entry often contained information about many different activities. It was therefore challenging to identify from the logbooks retrospectively when specific parameters of the functioning of the Riverlab were changed during previous interventions. Even though we did not implement it ourselves, an optimal strategy could include the possibility to register important settings as well as the start and end of an intervention day in a tabular format directly into the database. This has to remain simple, though, in order to avoid the addition of a time-consuming task during an already saturated fieldwork day.

Finally, the database has be to collected, archived and in a format that allows the unambiguous diffusion and usability of the data, with all relevant metadata.



## 6 Team structure, skills and organization

To run field laboratories over medium to long term (from several years to decades) it is necessary to dedicate available and
competent scientific and technical resources to ensure its proper functioning, the collection, analysis and archiving of a large
quantity of data. Even if the number of work hours required to produce these large number of analytical results is drastically
reduced by these field laboratories, the need for human resources remains important. The combination of skills required to run
such a field laboratory is also new and highly diverse. In this section, we aim at highlighting the organizational challenges
associated with such technological innovations and try to give recommendations to avoid neglecting these challenges.

### 6.1 Team organization

The organization of the Riverlab team members is mainly based on their skills and coordination, as many operations are
dependent on each other. The optimum functioning of the Riverlab require multiple and varied skills in the field of analytical
chemistry, sensor installation and calibration, electronics, hydraulics and others. All tasks and implications of each team
member should be clearly discussed and formalized. Therefore, it is necessary to establish a good communication mode of
interventions, breakdowns, adjustments, modifications etc. on the Riverlab, with an archive of the history of all operations.
For this purpose, an online notebook, accessible both in the Riverlab and in the offices and laboratories, is a well-adapted tool.
Furthermore, a precise intervention schedule, elaborated and shared between the different actors, is necessary.
The Riverlab requires interactions with many manufacturers, subcontractors and service providers. It is therefore necessary to
have an up-to-date directory of contacts with the technical relays, but also the commercial ones of these various structures.

### 6.2 Team skills

Field laboratories require therefore various and complementary skills. Obviously, as it is the case for any analytical laboratory,
it requires technical skills in chemical analysis, especially to run and control the data from the ICS. Technical skills in field
instruments for environmental monitoring are also precious to understand and to properly conduct the maintenance of the
various sensors. As highlighted in section 3, the supply of water to the analysers and instruments is also critical in the running
of the Riverlab and this requires skills related to the functioning of the pumping and filtration systems. As the field laboratory
is also a relatively heavy infrastructure, one needs also to apply rules related to health and safety, to control and fix the power
supply, etc. In addition, the remote nature of such field laboratories, and their purpose to produce high temporal resolution
data, requires skills in remote communication systems and in data management (information systems, databases…).
Synchronization issues, data curation, archiving and traceability of these processes require computer-assisted protocols.
Finally, communication and organizational skills are crucial regarding the amount of consumables and equipment and the
amount of contact persons.




In the case of the three Riverlabs, the strategy of sharing different expertise and skills in a team appeared to be successful and precious to solve technical issues, especially in this testing phase of the prototypes but required, again, strong communication skills. Sharing the maintenance of the Riverlab between several people was also necessary in order to continuously operate the field laboratories, which can includes on-call duty for weekends and holidays. Organization skills are therefore necessary for sharing the work and the information, and for using appropriated tools (e.g. shared calendar…).

## 7 Conclusion

The scientific perspectives in the fields of hydrology, geochemistry, aquatic ecology, and environmental geosciences have been suddenly and highly renewed with the development of field laboratories and other technologies enabling near continuous measurement of water quality. Acquiring funding for equipment and material can be considered relatively easy with several dedicated calls at national and European scales. However, the human and financial costs associated with such infrastructures are important and associated funding for running the equipment might be more difficult to identify and often poorly synchronized with the calls for other projects. Such an investment in financial and human resources involves many technical and organizational challenges.

In this technical note, we reviewed the critical aspects we experienced in the testing phase of three field laboratories for measuring in situ and at sub-hourly frequency the major element concentrations in stream waters. One of the main conclusions form our cross experiences is that a very large number of components and parameters needs to be adapted empirically to the local conditions (frost, drought and flooding frequency, flow intermittence, water turbidity, distance from offices...). This type of complex field laboratories is not one-fits-all systems that can be deployed anywhere without significant adjustments and in our experiences a settling period of two to three years was necessary. Furthermore, a non-negligible amount of diverse and detailed knowledge and human resources is paramount for the acquisition of reliable data. Specific technological skills are required to operate such a tool in the fields of hydraulics, filtration, electric, sensors maintenance, electronics, telecommunication, data treatment, data management. In addition, it is essential to validate data before it can be processed and interpreted. We believe that the critical steps identified here, and the solutions we recommend, can be transposed beyond our specific equipment from those tested by our three teams. This technical note is a practical guide to help to design the project of running these or similar field laboratories and then to acquire as fast as possible original and precious data sets. Promising results already came out of these experiences (Floury et al., 2017, 2024; Tonqui-Neira et al., 2020a; 2020b; 2021; Brekenfeld et al., 2023, submitted) and more will come. High-frequency geochemical data can be used to study processes in several disciplines (hydrology, biogeochemical cycles, ecology, microbiology, agronomy, etc.) and on several time scales (dial, event, seasonal, annual, etc.) with many avenues still to be explored.



**Funding**

The Critex Programme ANR-11-EQPX-0011 funded the Riverlabs and most of the costs associated with their running. The people who run the Riverlabs are staff from ORACLE, OHGE and AgrHyS Critical Zone Observatories (i.e. from CNRS and

INRAE). The research units UR HYCAR, UMR ITES/EOST, UMR SAS and UMR Geosciences Rennes contributed to the current costs especially: vehicle costs associated with regular travels on the site for maintenance, power supply, etc. The post-doctoral position of N.B. was co-founded by Region Bretagne, UMR SAS, INRAE (AQUA) and OZCAR-RI.

**Competing interests**

The contact author has declared that none of the authors has any competing interests

**Acknowledgment**

We thank Béatrice Trinkler (retired from INRAE Rennes), Laure Cordier (IPG Paris), and Sophie Ganglof (ITES Strasbourg) for their contribution to the running of the ionic chromatography systems during the first stages of the three prototypes, Gaelle Tallec (INRAE Antony), for her contributions to the design and adaptation of the first prototype,

Jérôme Gaillardet (IPG Paris) as the PI of the CRITEX project and lead of the Riverlab task.

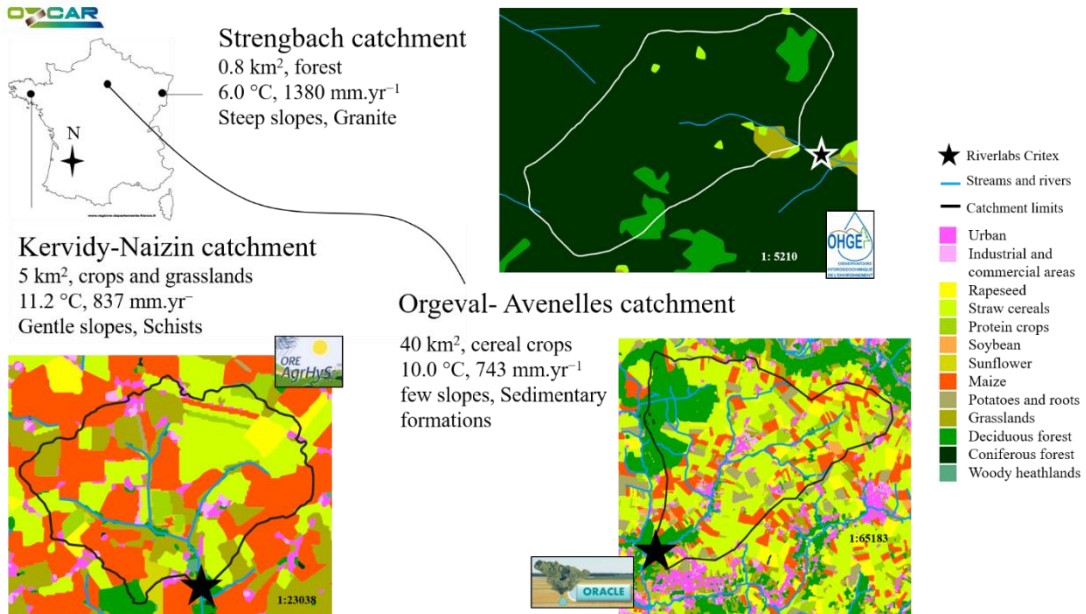

**Figure1: Location maps of the three Riverlabs in the Critical Zone Observatories of AgrHyS, Orgeval and OHGE. Sources: OSO (Inglada, Jordi, Vincent, Arthur, & Thierion, Vincent. 2019. Theia OSO Land Cover Map 2019 (Version 1) [Data set]. Zenodo.**



**https://doi.org/10.5281/zenodo.6538321**);                          **BD                                TOPAGE®,**
      (**https://www.sandre.eaufrance.fr/atlas/srv/fre/catalog.search#/metadata/a535c474-eca0-4295-880e-a95c8e633a51**, **28 April 2023**)

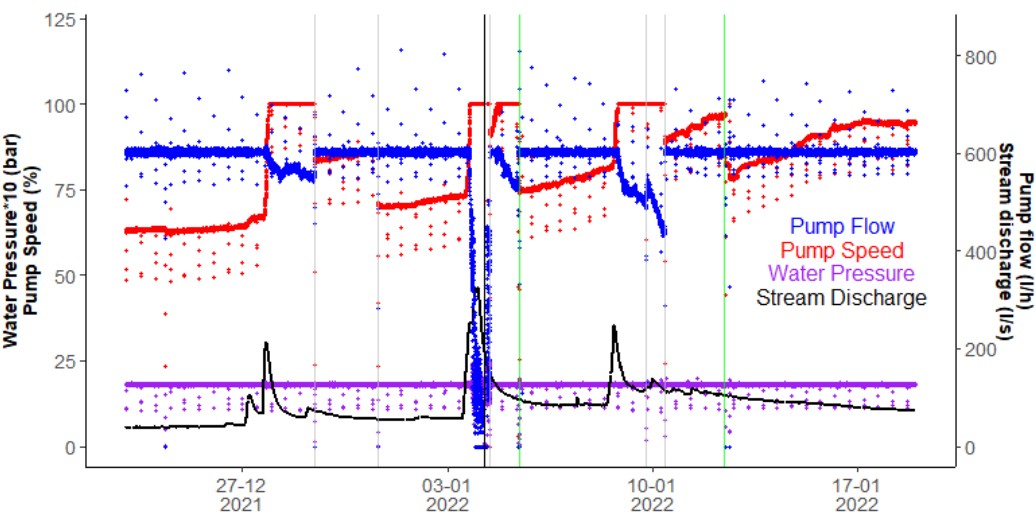


**Figure 2: Pump flow (blue, l/h), pump speed (red, %), water pressure (purple, bar) and stream discharge (black, l/s) at the Naizin catchment. A submersible pump was used: Grey bars indicate manual pump stops and re-starts, green bars indicated pump cleaning and black bars indicate an automatic pump stop due to instabilities. Note the periods of full pump speed (100%) during the storm events, when the targeted flow values are not reached, and the decreasing pump efficiency (red; increased pump speed necessary for**
**a given flow) over the course of those four weeks.**



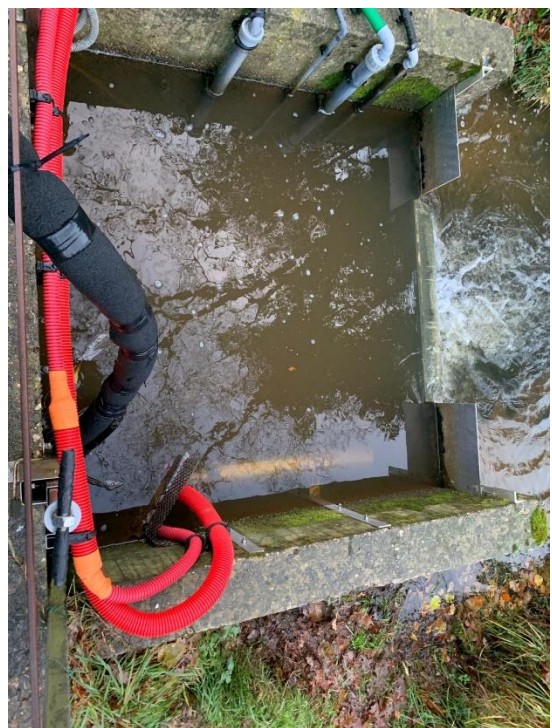

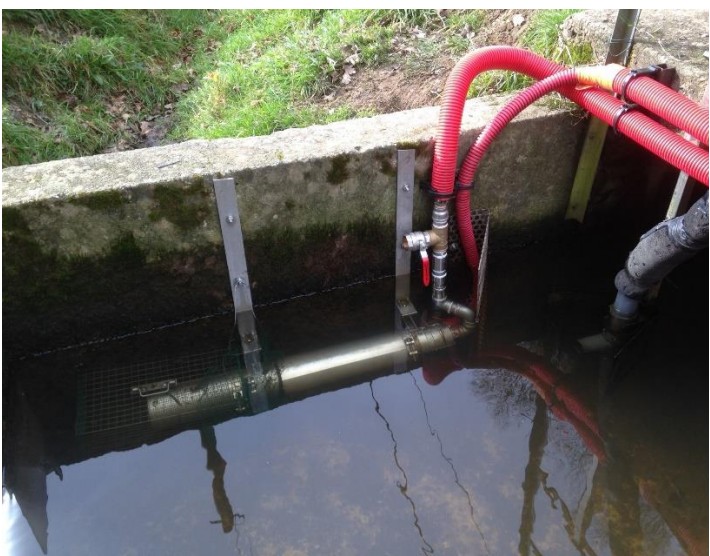

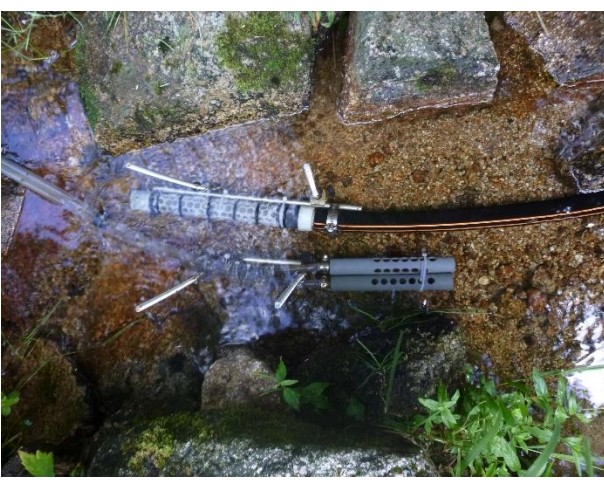

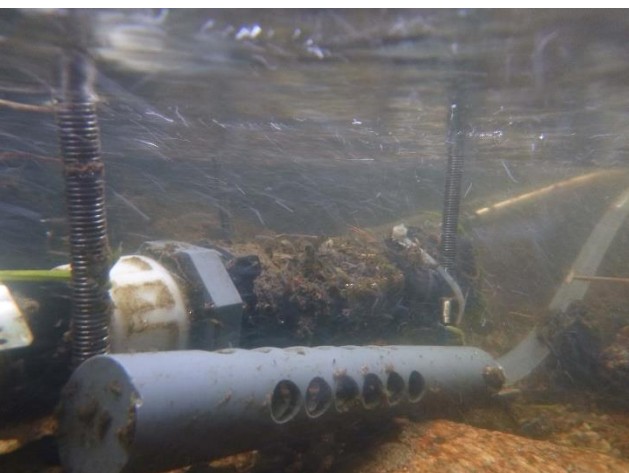

**Figure 3: Pictures of the strainers and baffles in the station of Naizin (top) and Strengbach (bottom) to prevent clogging of the pumps.**




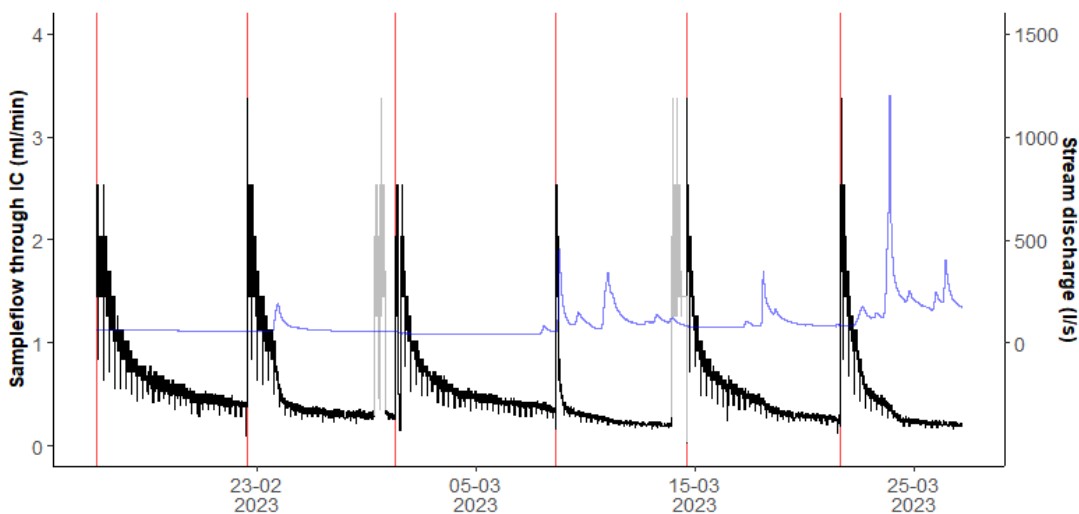

**Figure 4: Evolution of the flow at the output of the ionic chromatograph (IC) in black and stream discharge in blue, at the Naizin catchment. Red bars indicate a change of the spiral filter (0.22 μm) and grey periods are due to purging of the IC with ultra-pure water.**

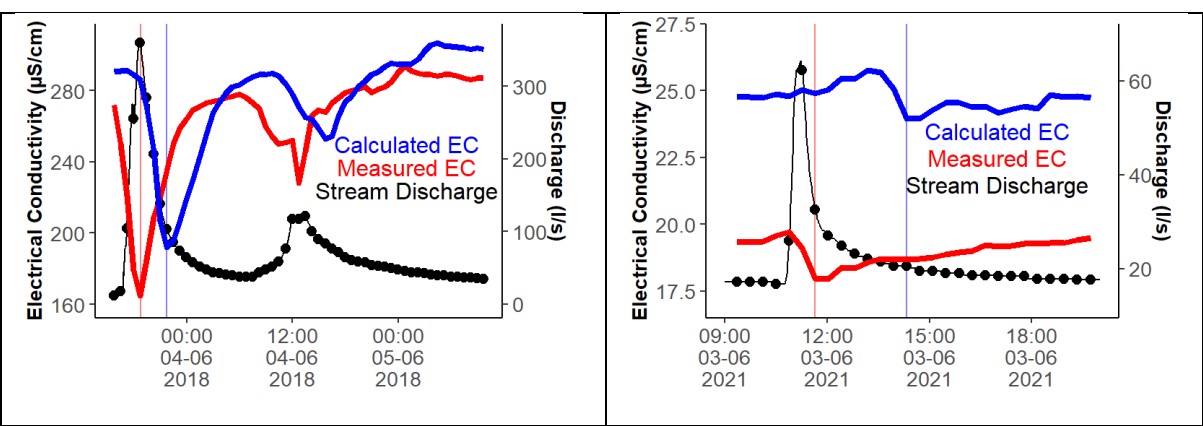

**Figure 5: Estimation of the transit time between the overflow tank and the Ion Chromatography system (IC) at the Naizin (left) and Strengbach (right) catchments. The transit time was calculated as the difference between the measured electrical conductivity in the overflow tank (red) and the calculated electrical conductivity computed from ion concentrations measured in the IC. The black dots along the hydrograph (black line) indicated the moments when a water sample was injected into the IC and the red and blue vertical bars indicate the timing of the minimum measured and calculated electrical conductivities, respectively. The time difference between these two bars amounts to 2h54 at Naizin (left) and to 2h41 at Strengbach (right) and can be used as a proxy for the transit time. The long transit time is primarily due to the clogging of the second filter (0.22 μm).**



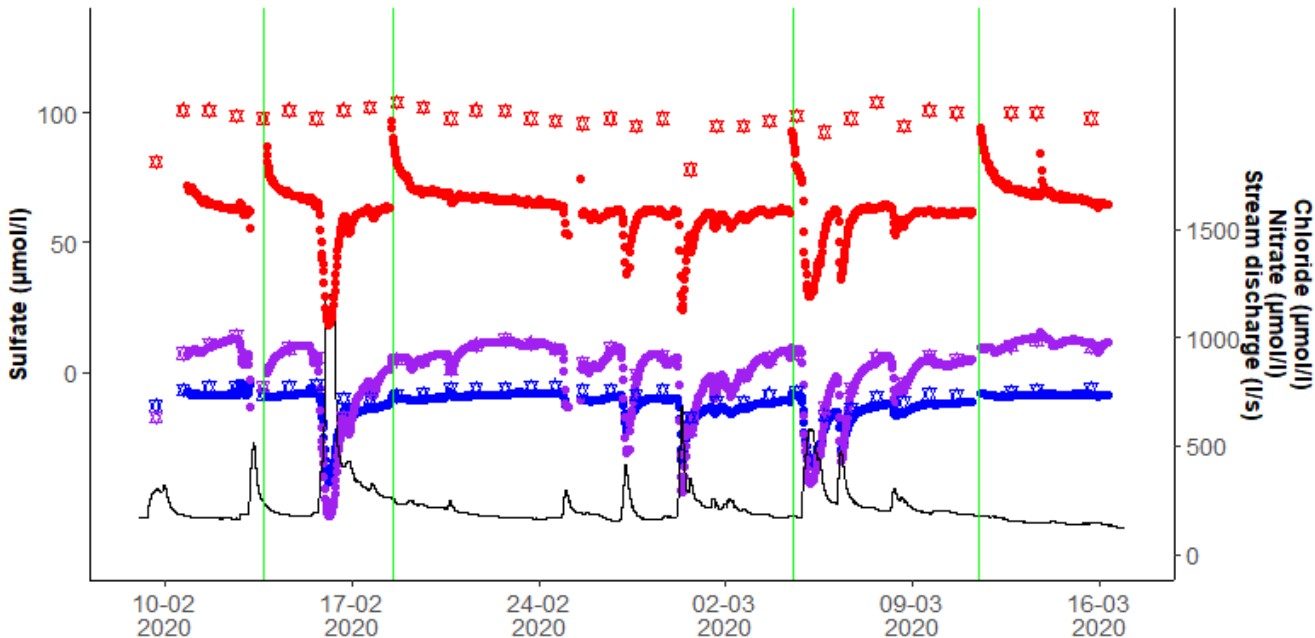

**Figure 6: Evolution of the sulfate (red), chloride (blue), nitrate (purple) concentrations and stream discharge (black) measured in the Riverlab at the Naizin catchment, green bars indicate a change of the spiral filter (0.22 µm). The points are concentrations measured in the Riverlab, whereas the stars are concentrations measured independently in our lab from daily grab samples from the stream.**

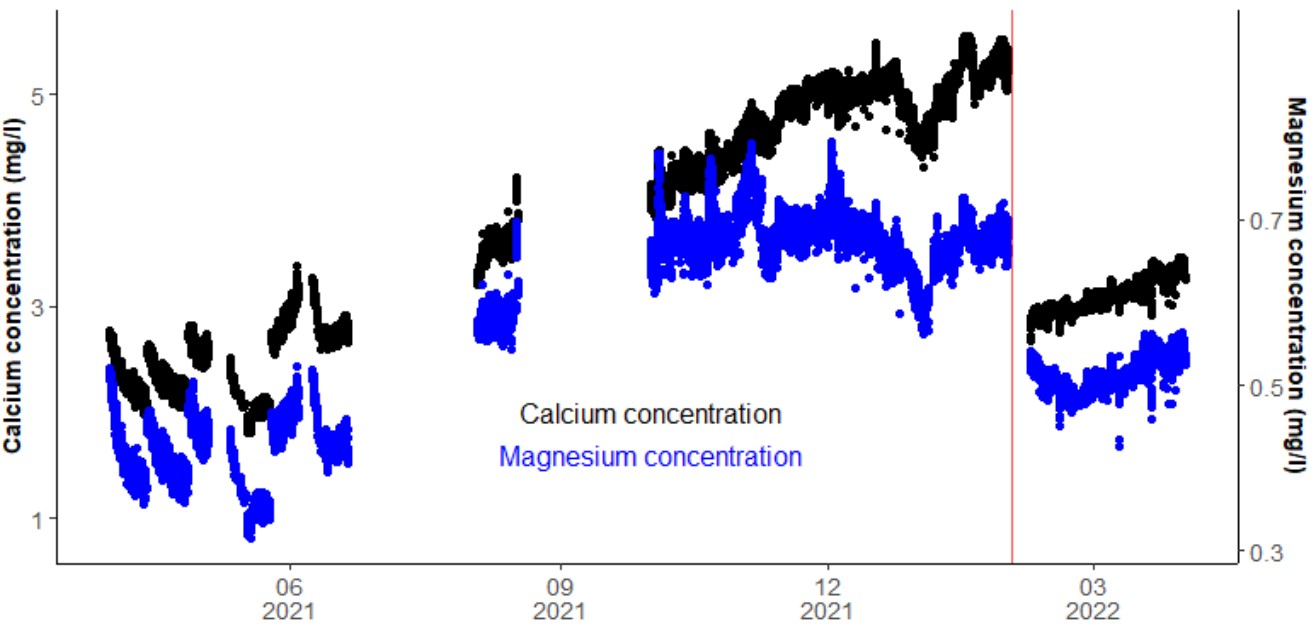



**Figure 7: A long-term and gradual contamination issue was observed in the three sites affecting cation analyses, specifically Calcium and Magnesium, here illustrated for Strengbach. According to our investigation, this issue is likely occurring either in the PEEK tubing used between the 0.22 μm filter and the ionic chromatography system which are several times longer than the tubing usually used in the laboratory or in the injection loops. The red, vertical bar indicate the date when the injection loop of the cations was changed.**

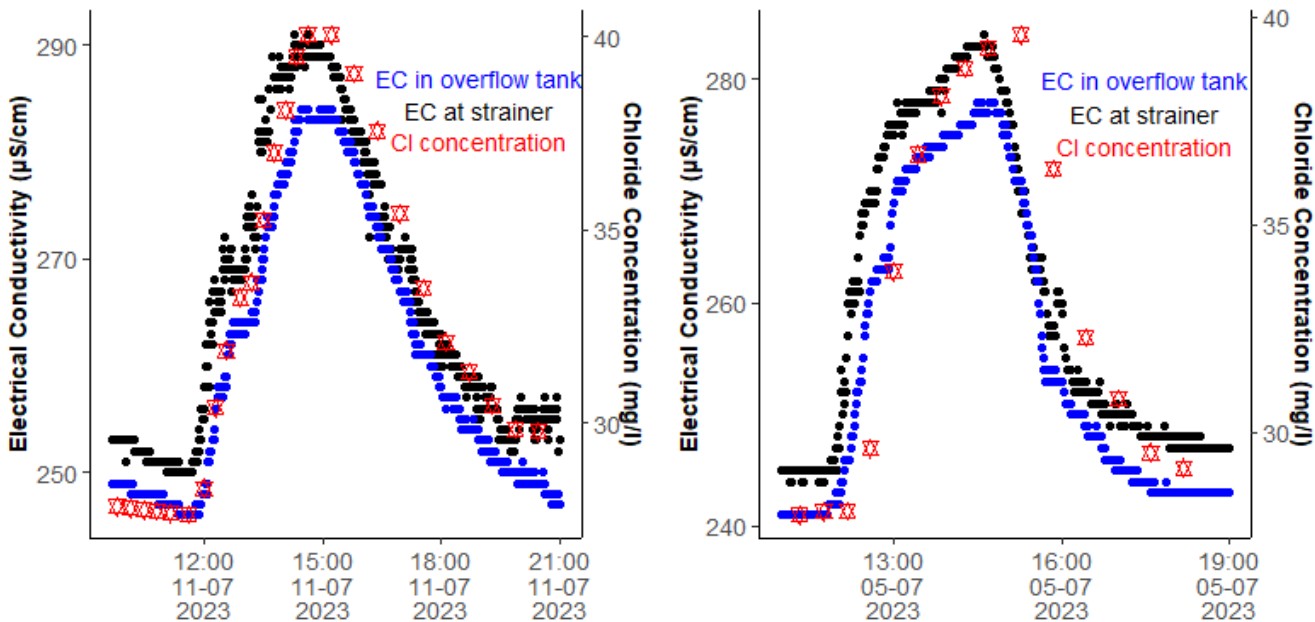

**Figure 8: Salt tracer test for the determination of the travel time within the Riverlab at Naizin with a clean (left) and a one-week-old (right) 0.22 μm spiral filter. For both injections, a salt solution was injected continuously 123 m upstream of the strainer for a period of around 70 minutes. The electrical conductivity was measured by a hand-held EC-meter next to the strainer (black) and in the overflow tank (blue). The chloride concentrations were measured by the ICS with manually shortened run times (15/20 minutes instead of 35 minutes). During both injections, the travel time between the strainer and the overflow tank was around 10 minutes. However, the travel time between the strainer and the ICS was around 10 minutes with a new filter (left) and around 35 minutes with an old filter (right).**

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
