# Peer review of "Technical Note on high-frequency, multi-elemental stream water monitoring: experiences, feedbacks, and suggestions from seven years of running three French field laboratories (Riverlabs)"

_EGUsphere, 2024_

## Author Comment (AC1)

[Figure]

Figure 9: Data management workflow of the Critex Riverlabs and associated challenges. QC/QA: quality control/quality assurance; O2: dissolved oxygen concentration, T°: water temperature, EC: electrical conductivity in water; [DOC]: dissolved organic carbon concentration

[Figure]

New Figure 1: Diagrams of the three Riverlab prototypes.

---

## Author Response (AR1)

**Reply to Comment from Referee 1 on EGUSPHERE-2024-902**

*Review of egusphere-2024-902: Technical Note on high frequency, multi-elemental stream water monitoring: experiences, feedbacks and suggestions from seven years of running three French field laboratories (Riverlabs).*

*This technical note (TN) is intended to review the challenges and considerations that arise in the placement, operation and maintenance of a specialized deployment designed to continuously analyze a flexible and rigorous suite of geochemical parameters in rivers semi-autonomously and in real time. I should clarify at the start that I also run a Riverlab facility, designed by the same company (Extralab). Ours was designed and deployed more recently (2021) and is the only one stationed in North America. From this rather unique perspective I can say that I appreciate and support the intended purpose of this TN, as it is indeed very challenging to appropriately run one of these facilities. However, I am concerned that there's sparingly little information offered in this TN that is relevant to the deployment we operate. Many of the issues described have been alleviated by updates to the Riverlab technology, while others are a result of the site-specific hydrogeochemical behavior of the Naizin and Strengbach watersheds rather than universally applicable guidance. Finally, there are multiple vital aspects of successful Riverlab operation that are overlooked or omitted here. For these reasons I do not believe that this TN serves its intended purpose of offering experience to benefit future deployments and I recommend that the manuscript be rewritten to reflect the current state of knowledge in working with such technology prior to further consideration for publication. I have detailed the major points this would require below, followed by in-line edits. These are offered in a spirit of collaboration and in hope of ultimately advancing rather than inhibiting this emerging technology.*

*Jennifer Druhan*

**Reply:** We thank you, Jennifer, for the time you spent and the care you gave in reviewing the manuscript of our TN. Your unique perspective is indeed relevant for evaluating this kind of TN. We are glad that you have faced fewer technical issues when running the University of Illinois Riverlab. We were disappointed that the information in the TN was not very relevant for your facility. Nevertheless, we still believe that it would be relevant and useful for others in the environmental science community. In our replies that follow, we explain why and provide our best arguments. We detail below how we revised the manuscript to address your comments, which undoubtedly improved the document. We thank you for your spirit of collaboration; we share the desire to advance this technology. It is precisely this motivation that led us to submit this TN, since sharing our experience with the wider scientific community could help capitalize on and accelerate progress.

*Major edits:*

1. *Current state of methodology: The final sentence of the abstract states "These considerations will same time, improve performance and ensure continuous field monitoring". It is necessary that the paper accomplish these tasks. The challenge faced is that this TN is based on three Riverlab facilities which were deployed between 2015 and 2017. In the interim since, there has been considerable development on the part of the manufacturer to improve a variety of aspects in the overall system design, operation, automation, and management, which are unfortunately missing in this TN. While some of the experiences described in this document do still offer useful guidance, many of the problems discussed in detail here (pump design, flow and pressure optimization, strainer, and intake issues, etc.) have been alleviated or are at least mitigated with new approaches. Unfortunately, this means that the information provided*

*here is largely out of date, specific only to very dilute systems, and thus not constructed in a manner that offers universal guidance to future deployments.*

**Reply:** It is true that we describe our experience with the three Critex Riverlabs in the Introduction (Lines 58-61). We are aware that new versions of Riverlabs exist, such as at the University of Illinois, and it is good that some of the technical issues have been overcome. However, we do not think that this makes our experience out-of-date; instead, we argue that:

1) While some of the issues we address are somewhat specific to very dilute systems (e.g., mountain streams) or to small headwater catchments, the information remains relevant for research teams working at similar sites or with similar systems.

2) This TN is not restricted to those who use Riverlabs, but can be useful for research teams who would like to build their own field laboratory prototype or who use a laboratory sold by other manufacturers that may target other elements/analyses.

Although few studies detail the technical issues we experienced, we know that they are not specific to our sites. For example, Cassidy and Jordan (2022) visited the sites of several monitoring programs in northwestern Europe that collected high-frequency water-quality data (including Riverlabs). They identified that "Sites and programs visited in this study had various technical issues and problems related to specific equipment ranging from breakdowns of moving parts or parts exposed to heat and pressure, pipe and pump blockage, seal damage, and data signal transfer. Some issues were related to the dynamic nature of river systems exposed to short-duration, sediment laden storm waters and heavy debris damage to in-stream ancillary equipment."

2. *Missing factors: The TN almost entirely overlooks the challenges associated with data management and software development necessary to automate the Riverlab facilities and ensure preservation, analysis and monitoring of large volumes of data. I think this would be a much more useful and appropriate topic for section 6 than the current "Team structure".*

**Reply:** In our experience, the main problem was actually not associated with data management (the focus of section 5.3). We agree that these facilities produce such a large volume (and variety) of data that it becomes necessary to change procedures for controlling and qualifying data, which should rely more on automatic cleaning/checking algorithms. In our experience, however, the analytical equipment needs to have its results monitored (and to be repaired when necessary) by an experienced technician, regardless of the other tasks that can be automated using a robot (e.g., sampling, injecting reagents, measuring) or an algorithm (e.g., the tool in chromatograph software that integrates conductivity to estimate concentrations). Therefore, it was not possible to fully automate data acquisition and quality control, which led us to identify three main challenges for data management. We have produced a new figure (Figure 9, provided in supplement here) and have made the following modifications to section 5.3 to make the messages clearer:

"Producing long time series that are easily accessible and usable later often requires combining manual and automatic quality-control procedures. Data management thus includes several challenges, as highlighted for various high-frequency water-quality monitoring facilities (Cassidy and Jordan, 2022). We identified three main challenges associated with data management (Fig. 9). In our experience, detailed technical knowledge of effects of the components and their interactions on measurements is also required. The ion concentration time series from the ICS was validated regularly and manually by technical staff members, who were maintaining the ICS and were thus aware of any relevant malfunctions or inaccuracies. This procedure allowed us to remove non-validated measurements from the time series.

Challenge 1 (Fig. 9) is therefore related to compilation of the various data sets, especially if sensors and technologies need to be synchronized (e.g., time lags, injection versus continuous flow analysis). In other cases, specific routines in the functioning of the Riverlab influenced certain measurements during specific periods. Interpreting these additional variations correctly required detailed technical knowledge. For example, programmed flushing of the entire pumping cycle artificially increased and decreased the temperature in the overflow tank slightly during the flushing. This rapid temperature variation in turn influenced the temperature-corrected electrical conductivity due to a lag in the temperature adaptation of the conductivity sensor. Another challenge (Challenge 3, Fig. 9) is thus related to translating this technical knowledge into algorithms that will use non-targeted monitored variables (e.g. room temperature, pressures, pump speed and velocity) to curate and qualify the monitored variables targeted (e.g. water temperature, pH, conductivity and concentrations).

All interventions in the Riverlab, such as maintenance work, modifications and repairs, were recorded in electronic (and sometimes paper) logbooks, which all project participants could share and access. Depending on the recording system, however, these interventions were not directly connected to the database or in tabular format. Because this information had to be extracted manually from the logbooks, it was not possible to visualize logbook data during a time series, such as the days when a specific filter was replaced during a concentration time series. In addition, due to the many activities performed during a single field day at the Riverlab, an individual log entry often described many different activities. It was therefore challenging to identify from the logbooks when specific parameters of the functioning of the Riverlab had changed during previous interventions. The last challenge (Challenge 2, Fig. 9) is therefore to convert the information in the logbooks into datastreams, or what we called "metadatastreams", to highlight that these time series characterize one or more other time series rather than being variables of interest in themselves. Finally, the database must be collected, archived and in a format that allows for unambiguous dissemination and usability of the data, with all relevant metadata."

Finally, the need for specific data-analysis techniques for nearly continuous data sets is not specific to Riverlabs, and Bieroza et al. (2023) discuss this point extensively for high-frequency water-quality data (their section 3.6. "Combining High-Frequency Water Quality Measurements with Statistical and Modeling Tools")

Regarding the relevance of "team structure", in our experience, it was a critical point to consider. We refer again to Cassidy and Jordan (2022), whose review also identified human skills and resources as critical, as summarized in their section "Personnel". Therefore, we think that details about the skills, working rules, and specific constraints or duties associated with the technologies used in our Riverlab prototypes in section 6 are relevant to share.

3.  *Repetition and omission of existing literature: The extent to which this paper overlaps with the Floury et al. 2017 Hydrol. Earth Syst. Sci publication on the Orgeval Riverlab is concerning. This prior publication detailed technical aspects of the operation including:*
    1.  *design, calibration, analytical performance;*
    2.  *accuracy and instrumental drift of the chromatography system;*
    3.  *whole-system precision and testing of cross-contamination using a salt test which appears to be the basis of the approach described here;*
    4.  *laboratory analysis to confirm reproducibility of major ion concentrations;*
    5.  *a detailed discussion of what is gained through high sampling frequency and improved analytical precision.*

*In many ways this earlier publication already offers much of what is presented in the current TN, in some cases more successfully. At the same time, the current TN omits any significant*

*use of the Orgeval facility or data, even though it is presented as one of the three deployments included in this note. Relatedly, I must point out that the authors have overlooked a recent publication by Wang et al. (2024) in STOTEN which used the Orgeval dataset, in combination with the earlier Plynlimon data and our own Riverlab deployment in Illinois. I was disappointed to see that such a relevant study, which uses data from one of the French deployments, was not listed among the publications used to demonstrate recent advancements based on these novel datasets. I recognize that the paper is from 2024, but it was out months before the Floury et al. (2024) Orgeval paper which the authors do cite, and both papers included Dr. Floury as a coauthor. Given that Dr. Floury is also coauthoring this TN, the omission is rather blatant. Finally, the TN does reference a submitted manuscript by the lead author which is not yet through the review process. This should be removed until it is appropriate to cite as a published study.*

**Reply:** Floury et al. (2017) was a proof-of-concept of the Riverlab as written in there; therefore, it was based on only one year of analysis of the Orgeval CZO. We respectfully disagree that the overlap of our TN with their article is concerning. We did repeat some of the context for the sake of clarity and cite Floury et al. (2017) for further details when relevant. In the TN, however, we share an experience of deploying the Riverlab at other sites that are smaller streams, with different chemistry and chemistry dynamics, especially different storm-event dynamics and over longer periods (seven years in total). The TN presents many vital technical aspects that are not mentioned by Floury et al. (2017), such as pumping, filtration, synchronization, and long-term effects. Although some of these aspects do not apply to the University of Illinois facility, we think that they remain relevant for other sites.

Regarding the article by Wang et al. (2024), we thank you for the suggestion, and we have now cited it to illustrate the scientific perspectives of such data sets:

- In the abstract we added information (Line 19-20):

"This trend should likely persist in the future as the technologies are still improving. Here, we share our experiences of running three French field laboratories (called Riverlabs) over seven years, a new version of the prototype deployed in 2021 along a larger river in Illinois, USA is not presented here."

- In the Introduction, we modified lines 46-51 as follows:

"Bieroza et al. (2023) identified six topics that high-frequency water-quality monitoring helped to advance significantly, and several studies demonstrated the importance of high-frequency concentration data for estimating the element loads exported from streams at annual or inter-annual scales (e.g. Cassidy and Jordan, 2011; Skeffington et al., 2015; Chappell et al., 2017; Wang et al., 2024)."

- In the introduction, we added line 61 the explicit mention to the new version of the Riverlab:

"It has to be noted that a new version of the Riverlab has been manufactured and installed along the Sangamon River in Monticello (Upper Sangamon River Basin US Critical Zone Observatory), Illinois, USA  in July 2021. This Riverlab designed to be deployed along a river draining 1500 km$^2$ catchment (Wang et al., 2024), much larger than the three catchments we present. Some of the technical issues we present in this technical note have been solved in this new version, and some are likely to be scale specific and therefore different between the the Critex and the Monticello cases."

- In the Conclusion, we added Wang et al. (2024) to the list of studies that are based on Riverlab data (lines 528-529). In this list, we also cite Brekenfeld et al. (in review) because the preprint is available on the EGUsphere. Of course, if this TN is accepted for publication but Brekenfeld et al. (in review) is not (yet), we would remove the citation.

*In-line edits:*

*L28: what is meant by "section"?*

**Reply:** This is the river cross section where water is sampled: water depth, stream width, etc. We suggest replacing "section" by "local river geometry".

*L34: "over the last two decades" but the oldest citation is from 2012*

**Reply**: Indeed, high-frequency water quality measurement techniques began to become a topic of interest for our research communities in the 2000s, in particular with **Kirchner's invited commentary in 2004 (https://doi.org/10.1002/hyp.5537),** which is now a historical reference and **which we will add to the list line 34**. These techniques have seen a real boom in their implementation since the 2010s as illustrated in Bieroza et al. (2023, Figure 1) by the number of articles.

*L36: "with the conviction that concentration data sets with increased resolution will be the key of advancing environmental sciences" I think this is a rather vague appraisal of the relevant Kirchner papers, which develop the need for high resolution chemical records based on frequency analysis. This is not a 'conviction' this is a quantitative analysis.*

**Reply:** We propose to rephrase as: "Over the last two decades, monitoring of water quality parameters at high temporal frequencies has strongly developed based on various technologies (Wade et al., 2012; Rode et al., 2016b; van Geer et al., 2016; Bieroza et al., 2023) with the motivation of advancing environmental sciences (Kirchner et al., 2004; Kirchner et al., 2023)"

*L38: what is a "riverbank side analyzer"?*

**Reply:** We refer here to chemistry analyzers for use at "bank-side" (Cassidy and Jordan, 2022), Chappell et al. (2017) described them as "technological advances that permit sampling and rapid chemical analysis on stream banks". We suggest rephrasing as: "Among these technologies, the emerging field laboratories, so far mainly used for surface waters, are running on the bank side of streams or rivers."

*L38: "can be qualified as the most sophisticated technologies" this is subjective and, in my opinion, incorrect. There are tremendous advances now occurring in a variety of portable and field-deployable riverine geochemical analysis that could be considered "more advanced" than a Riverlab.*

**Reply:** We propose the following modification: "Among these technologies, the emerging field laboratories, so far mainly used for surface waters, are running on the bank side of streams or rivers. Such wet-chemistry analyzers have often more requirements than optical or other in-stream sensors in terms of filtration and power supply."

*L39-40: "avoid some disadvantages…" I have to disagree here as well I don't think sample storage or delay in analysis are the major issues field-deployed real-time analytical capabilities circumvent.*

**Reply:** The storage and delay of analysis can be at major stake for several elements such as $PO_4^{2-}$ or $NH_4^+$, which are not the focus of the Riverlab prototype in our experiences though. We propose to modify the sentence as: "They consist of running chemical analytical instruments in the field, relaxing the constraints of travelling for sample collection, sample storage and related to the delay between sampling and analysis."

*L44: for the use in*

**Reply:** we suggest changing for "developed specifically for the research sector".

*L46-51: I'm failing to see how a summary / reiteration of the major conclusions of the Bieroza paper is relevant to the current scope and purpose of this technical note.*

**Reply:** We cut the sentence after "Bieroza et al. (2023) identified six topics where significant advancements were achieved thanks to high-frequency water quality monitoring."

*L54: it is very concerning that this statement appears to omit consideration of Floury et al. 2017 paper "The potentiometric Symphony: New progress in the high-frequency acquisition of stream chemical data", which offers extensive documentation of the technical and operational aspects of the Orgeval Riverlab deployment, to the point that aspects of the current TN may be repetitive in respect to this earlier publication.*

**Reply:** We cite the article of Floury et al. (2017) extensively in the TN lines 46, 99 and 528 and we fully acknowledge the importance of this proof of concept for the River Lab prototype but we precisely felt that repeating same aspects of this previous article in the submitted TN was not relevant. Floury et al. (2017) present the instrumentation, the data collected and analytical results from several performance tests over 1 to 2 days whereas the submitted TN deals with practicalities, failures and successes we experienced in relation with some precise technical choices related to instrumentation, its assemblages, and with a longer period of deployment.

*L55: 'urging' is not the correct word here.*

**Reply:** We removed the word.

*L66: "we usually illustrate encountered issues with illustrations" please correct this statement it's not appropriately constructed.*

**Reply:** We corrected as "we first illustrate encountered issues".

*L76: "and by a weathered layer" -- I'm not sure what this means.*

**Reply:** We refer to the superficial part of the bedrock which has experienced weathering processes, the "weathered zone" as defined in Ayraud et al. (2008, Fig. 8). https://doi.org/10.1016/j.apgeochem.2008.06.001

*L80: The description of Naizin should include a clear explanation of the extent to which this stream is ephemeral.*

**Reply:** We added the following sentence to explain: "The stream dries out almost every summer for a period up to 4 months"

*L98-101: This is really confusing. Either the three Riverlabs are "almost identical" or the latter two have "higher similarity". I'm pointing this out specifically because the remainder of the TN is so heavily focused on the Naizin and Strengbach systems while Orgeval is almost entirely omitted. If this is the reason why, then perhaps it is more appropriate to remove Orgeval entirely given the existing Floury et al. 2017 publication already describes a lot of what's to come.*

**Reply:** The three Riverlabs are **almost identical** but the Orgeval prototype was the first deployment and is therefore **slightly different** from the two other Riverlabs installed at the same time in the Strengbach and Naizin two years later. We think it is relevant to keep the three cases because 1) most of the issues observed in the two other prototypes were also experienced in the Orgeval case, and 2) when an issue has not been experienced in Orgeval it is interesting to highlight it as for example the absence of sulfate contamination (lines 247-249). We tried to clarify the first sentence L98/99 rephrasing as: "The three Riverlabs have a similar design and functioning (for further details about the first Riverlab see (Floury et al., 2017))". and deleting the last part of the second sentence L100/101.

*L107: I think it's important to mention here that this is not the only design option for the pump configuration.*

**Reply:** We use the section 3.1.1 to present and discuss the different pump configurations (including the option of a submersible pump). In order to make it clear, that we describe the pump configuration of the three French Riverlabs in this section (2.2), we added the following sentence (L 103): "In the following paragraphs, we describe the design and functioning of these three French Riverlabs. Note that in the US prototype, a specific set-up designed for the Sangamon River station is described in the supplements of Wang et al. (2024)."

*L108: missing turbidity sensor?*

**Reply:** It is described lines 110-111

*L139: "small fishes"?*

**Reply:** We deleted ""

*L154: This is just a consequence of the choice of which submersible pump one deploys. There are plenty that are designed to handle mud.*

**Reply:** It is true that there are many different types of submersible pumps, of which some might be able to handle different types of mud, debris etc. However, a pump that handles mud and sediments might require, perhaps, a more sophisticated or robust filtration system in the Riverlab. Furthermore, the stream geometry must be large enough to host other types of pumps, such as lifting pumps. We therefore highlight in the next section (3.1.2) that the pump type needs to be selected carefully, depending on these constrains.

In order to mention explicitly the option of a lifting pump, we added the following sentence (L193): "In hindsight, we acknowledge that a submersible lifting pump might had been more adequate than a borehole pump to handle fine organic and mineral particles. However, lifting pumps often require a deeper stream than borehole pumps and are therefore not an option in small streams."

*L171: The information provided here is outdated: These problems associated with variable frequency have been largely dealt with in most current PID offered by the manufacturer.*

**Reply:** This is very good indeed, the reason for listing in this TN all the issues we have met, including the fortunately numerous ones that have been solved, is that we believe that records of past issues are likely to facilitate and faster the resolution of future issues.

*L184-185: "… the absolute parameter values might not be very informative for other users, which use different PID systems". I think this is specifically the sort of issue that a rewriting of the TN should seek to alleviate.*

**Reply:** Indeed, we suggest suppressing these two sentences L181-185

*L204: this appears to be a site-specific design flaw, rather than universally helpful guidance.*

**Reply:** The differences between the site configurations provide here an overview of possible options with their advantages and constrains.

*L206-210: My understanding is that the pump is stopped and operations are ceased at Strengbach during prolonged freezes. This seems misleading?*

**Reply:** During the period, when the Riverlab at Strengbach was fully functional and the data exploitable (since autumn 2020), the operations were not stopped during the winter months due to prolonged freezes.

*L216: this would imply that standard protocols for collection of field samples and subsequent analysis in a laboratory, many of which involve filtration, are biased?*

**Reply:** The standard protocols do involve a relatively small volume of water that is filtered which limit the bias. However, filtering a large volume of water with the same filter that will be progressively clogged may influence the concentrations determined in the filtered water (likely due to adsorption of elements on particles retained by the filter). Considering a flow of filtered water between 0.03 and 0.18 l/h (Line 229) and a weekly change of the filter, the volume filtered by the filter just before its change varied between 5 and 30 l  that is much larger than the volumes usually associated with grab sampling.

To avoid confusion we propose to rephrase as: "However, filtration might influence the analysis when involving large volume of filtered water (Horowitz et al 1992)".

*L224: what exactly constitutes 'heavy fouling'? this seems quite arbitrary*

**Reply:** We replaced the sentence as: "However, during high-flow periods, we observed fast fouling of the tubing and the analytical equipment, which would have required a monthly or even bi-weekly cleaning or replacement of the entire tubing system."

*L233-235: We simply change the filter twice a week. This doesn't appear to be a critical issue*

**Reply:** Twice a week is not an intervention frequency possible for all sites, moreover as illustrated in figure 6, the clogging occurs very fast after filter change. We changed the conclusion to mention explicitly line 529 that "Several issues have been solved and some issues we experienced did not occur in the US deployment (Wang et al., 2024) where the instrumented station drained a larger area and is characterized by higher volume and fluxes

of water." This strengthen the idea that the technologies are improving fast and that new results come.

*L255: this seems like a design issue specific to this deployment. I have to say a lot of this section seems like issues specific to this first set of deployments which have been addressed since. It's thus hard to see how this serves as universal guidance for future deployments*

**Reply:** This issue is likely to occur when background concentrations in the water are very low, therefore we argue that without being universal this issue is not site specific but might apply to other sites with low background concentrations as in many mountainous, non-calcareous streams.

*L265-268: this is nonsensical. Of course the concentrations are different after filtration the particulates have been removed*

**Reply:** Filtration at 0.22 µm of water that is already filtered at 0.5 µm, is not supposed to affect the sulfate concentration that is supposed to be truly dissolved and not retained by the particles.

*L279: This 'crucial' statement is entirely empirical. Direct evidence should be provided that calibration curves are undermined after tubing changes, else this should be removed*

**Reply:** We provide in Figure 7 the data that illustrate the strong drop in the Ca and Mg concentrations after having replaced the PEEK tubing.

*L287: this is a simple process. I don't see why it's being so heavily emphasized here.*

**Reply:** First, the Figure 5 illustrates the process might not be as straightforward as it appears, because of the time lag between the different instruments and sensors. Also, physical and chemical sensors record an averaged value by minute while the ionic chromatography analyses a volume of water sampled during a few seconds with new volume injection every 30 or 40 minutes. The aggregation of both data sets can be done by averaging the variables at highest resolution over the lowest resolution time step or by subsampling the variables at highest resolution over the lowest resolution time. This might not be an issue if the concentrations vary at the scale of several hours or days but if one is interested in very fast variations, such as over several minutes at Strengbach during storm events, it becomes important.

*L295: again this is a site-specific design issue. The venting problem is not universal and the newer designs have circumvented this problem*

**Reply:** Following the recommendations from the two reviews, we propose to remove several details from the original manuscript and this paragraphs in particular would be one of them and transferred in the supplement information.

*L326: This is totally inappropriate – a TN cannot publish inferences from a conversation with a manufacturing engineer. This must be removed.*

**Reply:** We removed and rephrased as: "According to our investigations conducted with the help of the manufacturer company"

*L335: I have never seen any evidence that that a lower ion concentration causes faster wear of a pH probe. Please back this up with some references*

**Reply:** In the manuals of all pH probe manufacturers, the general advice is not to store the probe in demineralized or deionized water. In the case of our equipment, the reference is in "pH measurement. A practical guide to the installation, operation, maintenance and calibration of pH electrodes. Endress+Hauser. PU01052C/22/EN/01.14"

p. 10, in the chapter entitled "pH measurement FAQs", section entitled "What is the appropriate way to store a pH sensor when not in use?":

"The electrodes must be kept hydrated! Use electrolyte, pH4 buffer solution or tap water - never use deionised water"

*L338: this statement has been made multiple times through the manuscript with no examples. Please remove.*

**Reply:** The paragraph that follows this statement describes an example to illustrate this problem (340-346). In order to clarify this aspect, we shortened the paragraph (L338-346):

"In catchments with a fast changing water chemistry, it might be important to consider the differences in the way water is supplied to the different instruments to facilitate the process of synchronization of the acquired data. The DOC analyser in the Riverlabs at Naizin and Strengbach, for example, continuously extracts filtered water from a small overflow tank. In addition, the DOC analyser itself contains a reaction tube, which is replenished continuously. Its quasi-continuous measurements are an integration over several minutes to tens-of-minutes. This is in contrast to the ICS, where discrete measurements are based on the injection of a small sample volume at a precise and short moment of time. In addition, a variable and unequal transfer time from the stream to these different instruments adds another source of inconsistency. Therefore, combining these different types of data needs to be performed carefully with checking of the synchrony between measurements. "

*L340-346: Is this really a problem? It would seem the issue is related to what you are trying to measure and the frequency with which it changes. For example the 8 minute delay based on the salt test at Naizin (L357), you can't even see this in the IC measurement frequency*

**Reply:** Whether this is a problem or not largely depends on the questions one wants to answer with the dataset and on the flashiness of the system (Strengbach vs. Orgeval). We think it is important to mention this point so that future users of this or a similar field laboratory are aware of the potential problem this can pose. Once aware of it, they can evaluate it and decide by themselves how they want to proceed. See also our reply to the comment on L287.

*L369-375: Again, I'm failing to see whether this really matters – in comparison for example to where in the water column the intake system is situated. This seems quite specific to a small system*

**Reply:** It matters for flashy systems (with very fast rainfall-runoff response): we can expect this point being critical in systems dominated by surface flowpaths such as urban sites with high impervious area or sites where infiltration excess runoff is the dominant runoff generation process.

*L391-392: I find this entirely subjective, based on location of the deployment relative to those who can intervene. This is far too site specific to be offered as general guidance*

**Reply:** Certainly, this depends on the location of the deployment relative to those who can intervene and how flexible their time schedule is. To clarify this point, we added to the sentence in L391-392: "However, additional and spontaneous interventions might be required for solving some problems quickly."

*L420: exactly what 'waste' is this referring to?*

**Reply:** To clarify, we added to this sentence the following information: "In addition to the provision of the reagents for the different analytical instruments, it is very important to consider the handling of the produced chemical waste from these instruments (solvents, reagents etc.)."

*L434-447: this is entirely too specific to the manufacturer and specific technical support for these deployments. Other options exist, which circumvent these issues. I also don't think it's appropriate to publish a TN arguing for a costly maintenance contract with a corporation.*

**Reply:** Certainly, this paragraph reflects our experience from the three French Riverlabs and another manufacturer for ICS could be cheaper, nevertheless all companies offer maintenance contract with fees. We are not arguing for a costly maintenance contract. Instead, we describe, why and how we valued such a contract with the downside of an elevated cost.

*L471-472: This sentence should become an entire section of the TN. The complexity of software and data preservation necessary to run these systems is extraordinary, and in this sense the TN could offer a very helpful and universal guide for new deployments that is distict from the earlier Floury et al. (2017) paper*

**Reply:** We modified section 3 to include more aspects of data management as requested, please see details in reply to major comment 2.

*Section 6 should be removed. This is not useful or appropriate.*

**Reply:** As explained in the reply to general comment 2, in our experience and in other research team's experience in high-frequency water quality monitoring (Cassidy and Jordan, 2022; Bieroza et al., 2023), the human resources and skills are a critical point to consider. Therefore, we respectfully disagree that section is inappropriate.

*L512: hopefully beyond European scale?*

**Reply:** Yes of course, we are referring here to the funding opportunities and the possibility to associate with the investment in equipment, an investment in human resources. We rephrased as: "Acquiring funding for equipment and material can be considered relatively easy with several dedicated calls for research."

*L521: I really have to push back here: the idea of a 2-3 year "settling period" is absurd. The instrumentation is free of wear and in its best operational shape immediately after deployment. 2-3 years in many issues begin to arise. This is the most important period to collect data.*

**Reply:** Again, this reflects our experience from the three Riverlabs. We are not arguing that data should not be used and collected within the first 2 – 3 years, if all the instruments are running correctly. Of course, if everything works perfectly right from the beginning, then yes, the first 2 – 3 years are the most important to collect the data. We are not referring to a mandatory period, during which data collection is prohibited. Instead, we are providing, from

our experience, a realistic period between the installation and the full functioning of this type of complex field laboratories.

We could rephrase indeed as: "This type of complex field laboratories is not one-fits-all systems that can be deployed anywhere without significant adjustments, in our experiences a settling period of two to three years was necessary but, thanks to recent improvements, this period tends to be much shorter."

**References**

Jordan P and Cassidy R (2022) Perspectives on Water Quality Monitoring Approaches for Behavioral Change Research. *Front. Water* 4:917595. https://doi.org/10.3389/frwa.2022.917595

Skeffington, R. A., Halliday, S. J., Wade, A. J., Bowes, M. J., and Loewenthal, M. (2015) Using high-frequency water quality data to assess sampling strategies for the EU Water Framework Directive, Hydrol. Earth Syst. Sci., 19, 2491–2504, https://doi.org/10.5194/hess-19-2491-2015

Chappell, N.A., Jones, T.D., Tych W. (2017) Sampling frequency for water quality variables in streams: Systems analysis to quantify minimum monitoring rates, Water Research, Volume 123, Pages 49-57,https://doi.org/10.1016/j.watres.2017.06.047

Wang, J., Bouchez, J., Dolant, A., Floury, P., Stumpf, A.J., Bauer, E., Keefer, L., Gaillardet, J., Kumar, P., Druhan J.L. (2024) Sampling frequency, load estimation and the disproportionate effect of storms on solute mass flux in rivers, Science of The Total Environment, Volume 906, 167379, https://doi.org/10.1016/j.scitotenv.2023.167379

Brekenfeld, N., Cotel, S., Faucheux, M., Floury, P., Fourtet, C., Gaillardet, J., Guillon, S., Hamon, Y., Henine, H., Petitjean, P., Pierson-Wickmann, A.-C., Pierret, M.-C., and Fovet, O. (2023) Using high-frequency solute synchronies to determine simple two-end-member mixing in catchments during storm events, EGUsphere [preprint], https://doi.org/10.5194/egusphere-2023-2214

**Reply to Comment from Referee 2 on EGUSPHERE-2024-902**

General comments:

The manuscript presents valuable insights into high-frequency, multi-elemental stream water monitoring, drawing on seven years of experience with three French field laboratories (Riverlabs). The subject matter is highly relevant to the fields of catchment science and freshwater biogeochemistry. The authors offer a thorough overview of the technical and organizational aspects crucial for the successful implementation and long-term operation of such field laboratories. However, I still have some suggestions for the authors to improve this manuscript before it is published.

The technical note is based on experiences from Riverlab facilities deployed between 2015 and 2017. Since then, significant technological advancements have been made, which are not reflected in the manuscript. Many issues discussed, such as pump design, flow and pressure optimization, and intake challenges, have been resolved in newer systems. The manuscript should be updated to reflect these advancements to remain relevant.

The technical note lacks a discussion on the challenges of data management and the software development necessary for the automation and analysis of large data volumes. Including a detailed section on this topic would greatly enhance the usefulness of the work for future deployments.

Reply: We thank referee 2 for his/her careful review, we are grateful for these positive feedbacks on the content and topic of the technical note we submitted, and we thank reviewer 2 for the corrections he/she advised and the useful comments.

We suggest numerous modifications of our manuscript as detailed in reply to "specific comments" especially to address the three main aspects in relation with both reviewers' comments:

- Modifications for developing of the data management section,
- Manuscript enrichment with explicit references to the advancements made since our experience,
- Improving readability by removing some details or minor points

Specific comments:

The introduction section extensively cites literature, which reflects the research background. However, some parts appear overly lengthy and could be streamlined to emphasize key points. The authors can outline the advancements and updates in Riverlab technology since 2017 in the abstract to ensure readers understand the current state of the field.

In order to make clearer the advancements since our prototypes we suggest adding the following paragraph at the end of Line 61:

"It has to be noted that a new version of the Riverlab has been manufactured and installed along the Sangamon River in Monticello (Upper Sangamon River Basin US Critical Zone Observatory), Illinois, USA in July 2021. This last version of a Riverlab has been designed to be deployed along a river draining 1500 km² (Wang et al., 2024) that is much larger than the three cases tested during the Critex experience. Some of the technical issues we present in this technical note have been solved in this new version, and some are likely to be scale specific and therefore different between the cases of the Critex Riverlabs and the Monticello case. "

We also added some supplementary references lines 46-51 as follows:

"Bieroza et al. (2023) identified six topics that high-frequency water-quality monitoring helped to advance significantly, and several studies demonstrated the importance of high-frequency concentration data for estimating the element loads exported from streams at annual or inter-annual scales (e.g. Cassidy and Jordan, 2011; Skeffington et al., 2015; Chappell et al., 2017; **Wang et al., 2024**)."

We modified consistently the abstract to harmonize the information content (Line 19-20):

"This trend should likely persist in the future as the technologies are still improving. Here, we share our experiences of running three French field laboratories (called Riverlabs) over seven years, a new version of the prototype deployed in 2021 along a larger river in Illinois, USA is not presented here."

The methods section contains overly detailed operational descriptions. Consider moving some minor technical details to appendices to enhance the readability of the main text, and provide updated information on current methodologies, particularly regarding data management and software development.

 Reply

- We identified several minor details to remove and transfer in appendix (in particular: paragraphs related to the variable frequency drive in section 3.1.1, to the freezing winter temperatures in section 3.1.2., and to the ventilation system in section 4.1.)
- We suggest adding several explicit references to the prototype designed for the station located in the US (in abstract, introduction, section 2.2. and conclusion, see please our reply to major comment 3 from J. Druhan)
- We suggest additional information in section 5.3 to develop the aspects related to data management and software:

"Producing long time series that are easily accessible and usable later often requires combining manual and automatic quality-control procedures. Data management thus includes several challenges, as highlighted for various high-frequency water-quality monitoring facilities (Cassidy and Jordan, 2022). We identified three main challenges associated with data management (Fig. 9). In our

experience, detailed technical knowledge of effects of the components and their interactions on measurements is also required. The ion concentration time series from the ICS was validated regularly and manually by technical staff members, who were maintaining the ICS and were thus aware of any relevant malfunctions or inaccuracies. This procedure allowed us to remove non-validated measurements from the time series. Challenge 1 (Fig. 9) is therefore related to compilation of the various data sets, especially if sensors and technologies need to be synchronized (e.g., time lags, injection versus continuous flow analysis). In other cases, specific routines in the functioning of the Riverlab influenced certain measurements during specific periods. Interpreting these additional variations correctly required detailed technical knowledge. For example, programmed flushing of the entire pumping cycle artificially increased and decreased the temperature in the overflow tank slightly during the flushing. This rapid temperature variation in turn influenced the temperature-corrected electrical conductivity due to a lag in the temperature adaptation of the conductivity sensor. Another challenge (Challenge 3, Fig. 9) is thus related to translating this technical knowledge into algorithms that will use non-targeted monitored variables (e.g. room temperature, pressures, pump speed and velocity) to curate and qualify the monitored variables targeted (e.g. water temperature, pH, conductivity and concentrations).

All interventions in the Riverlab, such as maintenance work, modifications and repairs, were recorded in electronic (and sometimes paper) logbooks, which all project participants could share and access. Depending on the recording system, however, these interventions were not directly connected to the database or in tabular format. Because this information had to be extracted manually from the logbooks, it was not possible to visualize logbook data during a time series, such as the days when a specific filter was replaced during a concentration time series. In addition, due to the many activities performed during a single field day at the Riverlab, an individual log entry often described many different activities. It was therefore challenging to identify from the logbooks when specific parameters of the functioning of the Riverlab had changed during previous interventions. The last challenge (Challenge 2, Fig. 9) is therefore to convert the information in the logbooks into datastreams, or what we called "metadatastreams", to highlight that these time series characterize one or more other time series rather than being variables of interest in themselves.

Finally, the database must be collected, archived and in a format that allows for unambiguous dissemination and usability of the data, with all relevant metadata."

[Figure]

Figure 9: Data management workflow of the Critex Riverlabs and associated challenges. QC/QA: quality control/quality assurance; O2: dissolved oxygen concentration, T°: water temperature, EC: electrical conductivity in water; [DOC]: dissolved organic carbon concentration

For descriptions of laboratory setups and operational procedures, consider supplementing with flowcharts or diagrams to improve readability.

Reply: We propose the diagrams below as a new version of Figure 1, the location map (previous figure 1) will be moved to SI to keep a reasonable number of figures.

[Figure]

New Figure 1: Diagrams of the three Riverlab prototypes.

The technical details section redundantly mentions certain instrument models and functionalities. It is recommended to streamline these descriptions.

Reply:

We identified technical details in 3.1.1. (L146 – L148 and L165) and in 3.2.1. (L226 – L232) and moved them into section 2.2 in order to streamline these descriptions.

The discussion on "key aspects and challenges of the filtration system" could be more detailed. For example, specifically mention the effects of "high flow rates" and "long residence times" on analytical results.

Reply: We present two different effects on analytical results. One is illustrated in section 3.2 and relates to some unexpected effects on quantification of some elements (Figure 6, differences between red points and red stars). The other relates to the artificial lag that might occur due to a clogged filter. According to the experimental estimate presented in Figure 8, this delay is +25 min in one week (from 10 min with a new filter to 35 min). Figure 5 points out that this is time lag can be even larger.

In section 4.1, the description of interference between different devices can be split into two paragraphs, discussing the ventilation system's impact on water level measurement and the UPS system's effect on differential circuit breakers separately.

Reply: As requested by the reviewers in other comments we moved some of the information in SI such as the point related to the ventilation system

Some content repeats across different sections, such as interference issues between different equipment and sensors, mentioned in sections 4.1 and 4.3. These should be consolidated for a more concise and structured presentation.

Reply: Interference issues are not mentioned in section 4.3. However, the removal of several technical minor aspects we proposed (see reply to previous comments) should indeed help in making the manuscript more concise and to improve the presentation structure.

In section 4.3, provide detailed references for Extralab's solutions to support the stated content.

Reply: we propose to develop as: "For instance, the solution proposed by the Extralab company ("Protocol" function available in the ExtraLab® Dashboard, https://www.extralab-system.com/product, consultation on June 6, 2024) was to create filters …"

In section 5.2, discuss support and maintenance from manufacturers and suppliers in segmented paragraphs based on equipment type or issue type for better organization.

Reply: Thank you for the relevant suggestion, however we did not extend further this paragraph to keep it concise. This is in agreement with both reviewer's general request and indeed improved the readability of the manuscript. Therefore, it was difficult to

organize the description of maintenance issues by equipment type. We hope that table 1 already gives an overview of the requirements of each equipment.

Reassess the need for Section 6 on team structure. If it does not add significant value, consider replacing it with more pertinent content.

As argued in reply to the comment 2 of Jenny Druhan, we still think that section 6 is appropriate because in our experience and the literature mentions the team related aspects are critical.

In their review, Cassidy and Jordan (2022), identified human skills and resources as critical, as summarized in their section "Personnel". Bieroza et al. (2023) identified in the limitations of wet-chemistry analyzers: "challenges with sample filtration **(…), a high level of maintenance needed to prevent and troubleshoot sampling and analysis problems (such as clogging or freezing of sampling lines)**, and high power requirements."

A complete instrument such as the RiverLab requires skills and knowledge in a wide variety of fields, from electricity and remote data transmission, to hydraulics and process engineering. It requires as well travelling to the site and physical interventions on a regular basis. This is quite different from classical activity on an instrument in a laboratory or from classical use of in situ sensors as well.

Therefore, we claim that aspects related to the skills, working rules, and specific constraints or duties associated with the technologies used are relevant to share.

Global Applicability:

Line 34: change "has strongly developed" to "have strongly developed" for subject-verb agreement.

Line 216: Change "Horowitz et al 1992" to "Horowitz et al., 1992".

Line 149: "aspirating"?

Line 314: Change "disconnections errors" to "disconnection errors"

Reply to the four comments: Yes, we agree with all of these changes, thanks. However, instead of replacing "aspiring" by "aspirating", we replaced it with "drawing in".

Line 512: Provided a universal guidance is more applicable, not just specific to the French deployments.

Reply: As replied to the similar comment from J. Druhan, we rephrased as: "Acquiring funding for equipment and material can be considered relatively easy with several dedicated calls for research ."

**Citation**: https://doi.org/10.5194/egusphere-2024-902-RC2

References

Jordan P and Cassidy R (2022) Perspectives on Water Quality Monitoring Approaches for Behavioral Change Research. *Front. Water* 4:917595. https://www.frontiersin.org/journals/water/articles/10.3389/frwa.2022.917595/full

Skeffington, R. A., Halliday, S. J., Wade, A. J., Bowes, M. J., and Loewenthal, M. (2015) Using high-frequency water quality data to assess sampling strategies for the EU Water Framework Directive, Hydrol. Earth Syst. Sci., 19, 2491–2504, https://doi.org/10.5194/hess-19-2491-2015

Chappell, N.A., Jones, T.D., Tych W. (2017) Sampling frequency for water quality variables in streams: Systems analysis to quantify minimum monitoring rates, Water Research, Volume 123, Pages 49-57,https://doi.org/10.1016/j.watres.2017.06.047

Wang, J., Bouchez, J., Dolant, A., Floury, P., Stumpf, A.J., Bauer, E., Keefer, L., Gaillardet, J., Kumar, P., Druhan J.L. (2024) Sampling frequency, load estimation and the disproportionate effect of storms on solute mass flux in rivers, Science of The Total Environment, Volume 906, 167379, https://doi.org/10.1016/j.scitotenv.2023.167379

Brekenfeld, N., Cotel, S., Faucheux, M., Floury, P., Fourtet, C., Gaillardet, J., Guillon, S., Hamon, Y., Henine, H., Petitjean, P., Pierson-Wickmann, A.-C., Pierret, M.-C., and Fovet, O. (2023) Using high-frequency solute synchronies to determine simple two-end-member mixing in catchments during storm events, EGUsphere [preprint], https://doi.org/10.5194/egusphere-2023-2214

---

## Referee Report (RR1)

This paper provides a comprehensive overview of a seven-year high-frequency, multi-elemental stream water monitoring program (Riverlabs) in France. The study offers valuable technical and organizational insights that can benefit researchers and practitioners in high-frequency water quality monitoring. However, several areas require improvement to enhance clarity, depth, and generalizability.

**General Comments**

1. The abstract should explicitly highlight the innovative aspects of the Riverlab approach, particularly in sampling, filtration, and analysis, compared to existing high-frequency water quality monitoring methods. Additionally, the practical applications of the findings in improving water quality assessments across diverse hydrological settings should be clearly stated.
2. The introduction should provide a more comprehensive literature review on recent advancements in sensor technology, automated data processing, and filtration systems for high-frequency monitoring. Additionally, a stronger justification for the selection of the three catchments is needed, emphasizing their hydrological diversity and relevance. The introduction should conclude with a clear research hypothesis and expected contributions to better guide the reader.
3. Section 4.3 on Data Management and Software should include a more detailed discussion on challenges related to data interoperability, software compatibility, and synchronization when integrating multiple platforms. Furthermore, concerns regarding data security, such as access control, encryption, and backup strategies, should be addressed to ensure the protection of sensitive environmental data.
4. The study should provide a structured discussion on the generalizability of the Riverlab approach to different catchments, including arid regions, carbonate-dominated landscapes, and urbanized watersheds. Additionally, potential modifications required for scaling up the system to larger and more complex hydrological settings should be considered.
5. The conclusion should synthesize key technical and organizational lessons learned from the Riverlab experience and highlight their practical implications. Additionally, future research directions should be discussed, including the potential for next-generation sensors, artificial intelligence-driven data analysis, and remote sensing applications to further enhance high-frequency water quality monitoring.

**Specific Comments**

1. L111–117: Discuss the rationale behind selecting surface vs. submersible pumps, linking this choice to hydrological and sedimentological characteristics of each catchment.
2. L310–315: Provide more details on the detection methods for calcite precipitation in Orgeval and pH sensor degradation in Strengbach, along with their long-term impacts on data quality and measurement accuracy.

3. L376–378: Include quantitative data on the frequency and nature of maintenance issues, such as pump failures, sensor recalibrations, and filtration blockages, to better illustrate the operational demands and challenges of maintaining Riverlabs.
4. L505–510: Clarify the empirical adjustments made to system components based on field conditions. If no standardized protocols exist, discuss the feasibility of developing operational guidelines to optimize future monitoring deployments.